# Hardware-aligned Hierarchical Sparse Attention for Efficient Long-term Memory Access

**Xiang Hu[1], Jiaqi Leng[2], Jun Zhao[2], Kewei Tu[3*], Wei Wu[1*]**
[1]Ant Group, [2]Fudan University, [3]ShanghaiTech University
{aaron.hx, congyue.ww}@antgroup.com, tukw@shanghaitech.com
https://github.com/ant-research/long-context-modeling

## Abstract

A key advantage of Recurrent Neural Networks (RNNs) over Transformers is their linear computational and space complexity enables faster training and inference for long sequences. However, RNNs are fundamentally unable to randomly access historical context, and simply integrating attention mechanisms may undermine their efficiency advantages. To overcome this limitation, we propose **H**ierarchical **S**parse **A**ttention (HSA), a novel attention mechanism that enhances RNNs with long-range random access flexibility while preserving their merits in efficiency and length generalization. HSA divides inputs into chunks, selects the top-$k$ chunks and hierarchically aggregates information. The core innovation lies in learning token-to-chunk relevance based on fine-grained token-level information inside each chunk. This approach enhances the precision of chunk selection across both in-domain and out-of-domain context lengths. To make HSA efficient, we further introduce a hardware-aligned kernel design. By combining HSA with Mamba, we introduce RAMba, which achieves perfect accuracy in passkey retrieval across 64 million contexts despite pre-training on only 4K-length contexts, and significant improvements on various downstream tasks, with nearly constant memory footprint. These results show RAMba's huge potential in long-context modeling.

## 1 Introduction

The success of Large Language Models (LLMs) [1, 11, 59] has been largely driven by the Transformer architecture [60]. However, the quadratic computational and memory costs of self-attention make it inefficient for processing long sequences. Moreover, Transformers often struggle with inputs that exceed their pre-training length. These limitations have renewed interest in alternative architectures such as Recurrent Neural Networks (RNNs) [10, 24, 28, 33] that enable efficient, linear-time processing of sequential data while retaining a degree of extrapolation capability.

However, RNN-based models suffer from a critical limitation: the information bottleneck [58] caused by compressing variable-length contexts into fixed-dimensional representations. Unlike attention mechanisms, they lack random access to contextual information, which becomes especially problematic in tasks like passkey retrieval, where performance degrades as sequence length increases [62]. While augmenting RNNs with attention mechanisms can help mitigate this limitation, it introduces drawbacks such as poor length extrapolation, quadratic computational complexity, and substantial memory overhead during inference, ultimately undermining the original efficiency advantages of RNNs. Consequently, there remains no satisfactory RNN-based solution that can simultaneously achieve length generalization, random-access flexibility, and efficiency.

To address the trilemma, we propose a novel **H**ierarchical **S**parse **A**ttention (HSA) mechanism. Existing sparse attentions like NSA and MoBA [32, 70] typically divide sequences into chunks,

---

*Corresponding authors.

39th Conference on Neural Information Processing Systems (NeurIPS 2025).

allowing each token to attend to a concatenation of $k$ selected chunks, thus can potentially achieve both efficiency and random access. While this design offers a promising step toward resolving the trilemma, a closer examination reveals a critical weakness: these methods often suffer from inaccurate chunk selection, both within the training distribution and when generalizing to longer, out-of-domain contexts. As illustrated in Figure 1(A), the issue may stem from learning token-to-chunk relevance in a chunk-unaware way, relying on token-to-token gradients instead of chunk-level feedback.

Based on this insight, HSA introduces a two-stage hierarchical mechanism for the selected chunks to enable end-to-end learning of token-to-chunk relevance. As shown in Figure 1(B), in the first stage, it applies attention separately to the tokens within each chunk to capture chunk-level information. In the second stage, it fuses the chunk-level information by applying weighted summation using token-to-chunk weights. During the backpropagation pass, token-to-chunk weights are adjusted based on the contribution of the entire chunk to the next token prediction, achieving chunk-aware learning. Experimental results show that this chunk selection mechanism remains accurate even when context lengths exceed pre-training lengths by over 10,000 times in passkey retrieval. Since each token corresponds to different $k$ chunks, a naive implementation would require substantial memory.

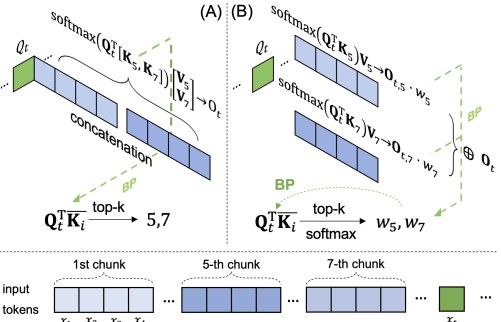

Figure 1: $\mathbf{K}_i, \mathbf{V}_i$ are the $i$-th chunk's key and value, with $\bar{\mathbf{K}}_i$ the mean pooling of $\mathbf{K}_i$. In (A), the chunk selection scores $\mathbf{Q}_t^\top \bar{\mathbf{K}}_i$ are learned from token-to-token interactions (*chunk-unaware*). In HSA (B), $\mathbf{Q}_t^\top \bar{\mathbf{K}}_i$ are guided by the feedback from the entire chunk (*chunk-aware*), with $\mathbf{O}_{t,i}$ the chunk-level information obtained from the $i$-th chunk by the $t$-th token.

Thus, we further propose a hardware-aligned algorithm to achieve efficient parallel computation.

With HSA, we propose **R**andom **A**ccess **M**am**ba** (RAMba), an extension of Mamba-2 [16] that incorporates HSA into specific layers. To maintain a constant memory footprint during inference, RAMba offloads the token-level key-value (KV) cache to CPU memory while retaining compact chunk-level representations on the GPU for efficient chunk selection. At each time step, only the KV cache of the selected chunks is loaded onto the GPU, ensuring efficient memory usage. To further minimize GPU-CPU memory swaps, RAMba leverages the hidden states derived from an intermediate layer as shared KV cache for all subsequent HSA layers, requiring just one chunk selection and swap per step. This architecture can simultaneously balance training and inference efficiency, length generalization, and long-range random access flexibility.

In our experiments, we compare RAMba with baselines such as Transformers, Mamba-2, and their variants with sliding window attention [14] and NSA [70], evaluating performance across long-range language modeling, downstream tasks, and efficiency. RAMba consistently outperforms the baselines in long-context modeling and downstream tasks while exhibiting exceptional length generalization. Notably, it is the first Mamba-based model to achieve perfect accuracy on a 64M context in the passkey retrieval task. In terms of efficiency, HSA is $3\times$ faster than NSA and $5$–$25\times$ faster than full attention for contexts of 16K tokens or more during the forward pass. Additionally, when memory offloading is enabled, RAMba maintains nearly constant memory usage. These results demonstrate RAMba's superior capability in long-text modeling. In summary, our contributions are threefold:

1. We propose HSA, a novel hierarchical attention mechanism paired with a hardware-efficient algorithm that simultaneously enables efficiency, length generalization, and flexible long-range random access.

2. Based on HSA, we introduce RAMba, which integrates the advantages of the attention mechanism into Mamba while maintaining a nearly constant memory footprint during inference.

3. We conducted comprehensive experiments on the length generalization of Mamba with various attention mechanisms. The results show that HSA excels in both performance and efficiency.

## 2 Related Works

**Sparse Attentions.** Sparse attention aims to reduce computational complexity by focusing on a limited number of tokens. For example, sliding window attention and its variants [9, 14, 41, 71] restrict computations to a fixed-size local window for each token. Such methods often sacrifice

the ability to capture long-range information. Clustering based approaches [30, 52, 61, 64] employ locality-sensitive hashing or K-Means for token clustering and perform attention with clusters. These methods often trade off efficiency for quality due to the limited accuracy of clustering. Cache eviction approaches [21, 31, 74] maintain constant memory costs by retaining only the most important tokens in the KV cache, but this limits the model's ability to access arbitrary contexts randomly. Combiner [50] utilizes a hierarchical attention mechanism by compressing tokens within chunks into single key-value pairs via max-pooling. However, this approach sacrifices the ability to capture token-level information inside the chunk, resulting in reduced accuracy. Recently, NSA [70] and MoBA [32] achieve sparsity by dividing a sequence into chunks and dynamically selecting relevant chunks for each token based on summed token-level attention scores. However, these approaches struggle with accurately identifying important chunks. Our work introduces a two-stage hierarchical attention mechanism with sparse chunk selection, achieving end-to-end learning of token-to-chunk relevance, while maintaining the ability to capture token-level information, resulting in attention flexibility, accuracy, and efficiency.

**RNN-base Language Models.** RNN-based language models have recently gained increased attention because their per-token inference cost remains constant as the sequence length grows. Linear Attention [28] shows that replacing softmax attention with kernel-based approximations allows Transformers to be reformulated as RNNs, achieving similar performance while benefiting from recurrent properties. Recent advancements in RNNs, such as RWKV series [42, 43], Mamba series [17, 23], GLA [66], HGRN series [46, 47], xLSTM [8] and others [67, 73], continue to push the boundaries of this approach. Our work builds on these RNN models and is in principle applicable across different RNN architectures. Meanwhile, other studies [18, 37, 51] explore hybrid architectures using RNNs for long-range memory and attention for short-range patterns. In contrast, our approach leverages RNNs to adaptively retain essential short-range information for next-token prediction, while sparse attention selectively accesses long-range context at arbitrary positions.

**Context Length Extrapolation.** The full attention mechanism struggles to generalize to contexts longer than those observed during pretraining [45], even with techniques like relative positional encoding [54] or attention weight scaling [63]. While methods such as Landmark Attention [35] and DCA [3] have been proposed to address this limitation, they still encounter a sharp rise in perplexity when extrapolating beyond a certain length, typically $32\times$. Recently, GCA [27] achieves perfect accuracy on 16M context lengths despite being pre-trained on only 4K lengths by integrating end-to-end retrieval within attention. However, GCA limits retrieval to once every $S$ (usually 64) tokens, reducing its flexibility. The two-stage attention mechanism in HSA is inspired by GCA, but achieving per-token retrieval through kernel optimization.

## 3    Methodology

From the psychological perspective, human memory primarily comprises two main systems: working memory [6], which temporarily stores and manipulates information with limited capacity, and long-term memory [5], which stores information indefinitely with virtually unlimited capacity. Inspired by this, RAMba applies Mamba to simulate working memory by compressing variable-length contexts into a finite representation for manipulation. Meanwhile, it uses HSA to model long-term memory as an extendable KV cache, enabling efficient retrieval and attention computation. In HSA, the key innovation lies in its two-stage hierarchical attention mechanism: first, weighting over tokens within a chunk, and then weighting over chunks. By offloading KV cache to CPU memory or disk, it can theoretically maintain unlimited memory. In the following sections, we elaborate the model architecture, HSA, kernel design, and optimizations for training and inference.

### 3.1    Model Architecture

As shown in Figure 2(a), RAMba contains $L$ Mamba layers, equally divided into upper and lower decoders, akin to previous works [27, 53]. Given the input sequence $\{x_1, ..., x_l\}$, their corresponding embeddings are fed into the lower decoder and the outputs are divided into $\lceil \frac{l}{S} \rceil$ chunks according to chunk size $S$. These chunked representations are then passed through a Transformer-based bi-directional encoder independently to form chunked memories, used for chunk selection and sparse attention in subsequent HSA layers. The upper decoder takes the outputs of the lower decoder as inputs, and alternates between one HSA layer and $G$ Mamba layers for $\frac{L}{2G}$ times. Between the

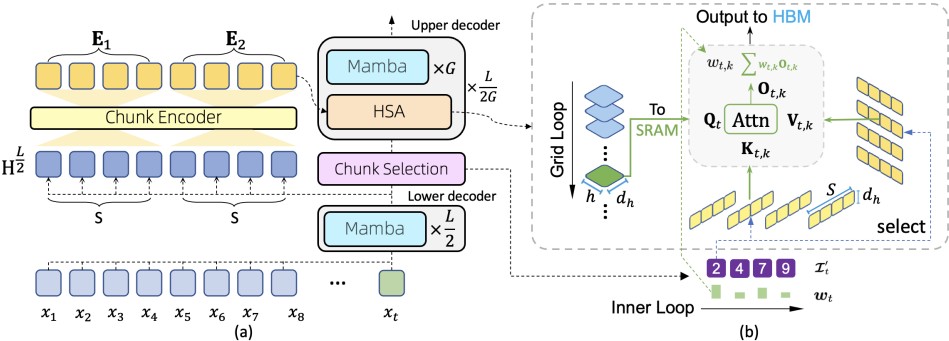

Figure 2: (a) Model architecture for RAMba. (b) Kernel design for HSA.

lower and upper decoders, a chunk selection layer selects the most relevant $k$ chunks based on the dot-product similarity between token and chunk representations at each decoding step. HSA layers capture long-range dependencies by attending to the selected chunks.

**Notations.** We focus on architectures like GQA [2], where each group of $h$ query heads shares a common KV head, and selects chunks independently. The hidden size $d$ is divided into $g$ groups, satisfying $g \times d_g = d$, with $d_g = h \times d_h$. Here, $d_g$ is the dimension of each group, and $d_h$ is the dimension per head. We denote the $m$-th layer output as $\mathbf{H}^m \in \mathbb{R}^{l \times d}$, with the $t$-step representation $\mathbf{H}^m_t \in \mathbb{R}^d$, and the encoded token representations of the $i$-th chunk as $\mathbf{E}_i \in \mathbb{R}^{S \times d}$. To simplify the notation, we use $\hat{\square}$ as a generic symbol to represent one of the groups, as each group behaves in the same way. For example, $\hat{\mathbf{H}}^m_t \in \mathbb{R}^{d_g}$ denotes one of the grouped representations.

**Chunk Selection.** At each time step $t$, a token uses the representation derived from the lower decoder to select top-$k$ past chunks. Each query group independently selects $K$ chunks, which are shared across the $h$ heads within the group. Formally, we have:

$$\mathbf{Q}^{slc}_t = \mathbf{W}^{slc}_Q \operatorname{norm}(\mathbf{H}^{\frac{L}{2}}_t), \quad \hat{s}_{t,i} = \begin{cases} \hat{\mathbf{Q}}^{slc\top}_t \hat{\mathbf{K}}^{slc}_i / \sqrt{d_g}, & i \leq \lfloor \frac{t}{S} \rfloor \\ -\infty, & i > \lfloor \frac{t}{S} \rfloor \end{cases}, \quad \hat{\boldsymbol{\mathcal{I}}}_t = \{i \mid \operatorname{rank}(\hat{s}_{t,i}) < K\},$$
$$\mathbf{K}^{slc}_i = \mathbf{W}^{slc}_K \bar{\mathbf{E}}_i$$

where $\operatorname{norm}$ is RMSNorm [72], $\mathbf{W}^{slc}_Q, \mathbf{W}^{slc}_K \in \mathbb{R}^{d \times d}$ conduct linear transformations, $\operatorname{rank}(\cdot)$ denotes the ranking position in descending order, $\bar{\mathbf{E}}_i \in \mathbb{R}^d$ is the mean-pooled representations of $\mathbf{E}_i$. $\mathbf{Q}^{slc}_t, \mathbf{K}^{slc}_i \in \mathbb{R}^d$ are representations used for chunk selection, with $\hat{\mathbf{Q}}^{slc}_t, \hat{\mathbf{K}}^{slc}_i \in \mathbb{R}^{d_g}$ as their grouped representations. For each group, $\hat{s}_{t,i}$ is the relevance score of $x_t$ to the $i$-th chunk, and $\hat{\boldsymbol{\mathcal{I}}}_t$ is the selected $K$ chunk indices for $x_t$.

### 3.1.1 Hierarchical Sparse Attention

**Chunk Weights.** To avoid the impact of position encoding on length generalization [29], we discard it and instead model the ordinal relationships of distances akin to the stick-breaking attention [55]. In the stick-breaking process, participants sequentially take a portion from a stick, with later participants dividing the remaining portion left by earlier ones, thus introducing a "most recent" bias. Let the total weights 1 serve as the stick and $w_{t,k}$ denote the weight assigned to the $k$-th chunk selected by the $t$-th token. For each group, we have:

$$\hat{\boldsymbol{\mathcal{I}}}'_t = \operatorname{sort}(\hat{\boldsymbol{\mathcal{I}}}_t), \quad \hat{w}_{t,k} = \hat{\beta}_{t,k} \prod_{i<k}(1 - \hat{\beta}_{t,i}) = \sigma(\hat{s}_{t,\hat{\boldsymbol{\mathcal{I}}}'_{t,k}}) \prod_{i<k}(1 - \sigma(\hat{s}_{t,\hat{\boldsymbol{\mathcal{I}}}'_{t,i}})),$$

where $\hat{\beta}_{t,k}$ is the proportion taken from the remaining attention, calculated via the sigmoid function $\sigma$. The $\operatorname{sort}(\cdot)$ function rearranges $\hat{\boldsymbol{\mathcal{I}}}_t$ in descending order to prioritize chunks closer to $x_t$ in attention allocation. $\hat{\boldsymbol{\mathcal{I}}}'_{t,k}$ denotes the the $k$-th value in $\hat{\boldsymbol{\mathcal{I}}}'_t$.

**Hierarchical Attention.** After obtaining the weights for each chunk, $x_t$ performs attention with tokens in each retrieved chunk separately. We denote the representation of information collected from the $k$-th selected chunk as $\mathbf{O}_{t,k} \in \mathbb{R}^{d_g}$. These representations are then fused using the chunk weights.

Formally, for the $l$-th HSA layer, we have:

$$\mathbf{Q}_t^l = \mathbf{W}_Q^l \,\mathrm{norm}(\mathbf{h}_t^{l-1}), \quad \mathbf{K}_i = \mathbf{W}_K \mathbf{E}_i, \quad \mathbf{V}_i = \mathbf{W}_V \mathbf{E}_i,$$

$$\underbrace{\hat{\mathbf{O}}_{t,k}^l = \mathrm{attn}(\hat{\mathbf{Q}}_t^l, \hat{\mathbf{K}}_{\hat{\mathcal{I}}_{t,k}'}, \hat{\mathbf{V}}_{\hat{\mathcal{I}}_{t,k}'})}_{\text{token-level attention for one group}}, \quad \underbrace{\hat{\mathbf{O}}_t^l = \sum_{k\,<K} \hat{w}_{t,k}\hat{\mathbf{O}}_{t,k}^l}_{\text{chunk-level attention for one group}}, \quad \underbrace{\mathbf{H}_t^{l+1} = \mathbf{H}_t^l + \mathbf{O}_t^l}_{\mathbf{O}_t^l \text{ is the concatenation of all groups}},$$

where $\mathbf{W}_Q^l \in \mathbb{R}^{d\times d}$, $\mathbf{W}_K, \mathbf{W}_V \in \mathbb{R}^{d\times(g\times d_h)}$ apply linear transform to obtain query, key, and value representations, with $h$ query head shares the same KV head. $\mathbf{K}_i, \mathbf{V}_i \in \mathbb{R}^{S\times(g\times d_h)}$ are the key and value corresponding to the $i$-th chunk, while $\hat{\mathbf{K}}_i, \hat{\mathbf{V}}_i \in \mathbb{R}^{S\times d_h}$ are their grouped representations, shared across all HSA layers. $\mathbf{Q}_t^l$ is the query representation of $x_t$ at the $l$-th layer.

**The key Difference with MoBA/NSA.** The primary distinction of HSA lies in the learnable chunk retrieval module. To illustrate the difference between existing sparse attentions, such as in MoBA [32] and MiniCPM4 [56], we provide a detailed analysis with specific examples in the Appendix A.

### 3.1.2 Kernel Design

In HSA, each token corresponds to a distinct set of $K$ chunks, which can lead to a substantial memory footprint in a naive implementation. Inspired by NSA, we address this issue by implementing hardware-aligned HSA kernels based on Triton [57]. As illustrated in Figure 2(b), each GPU thread loads the query representations for a single step along with the selected chunks' KV pairs for attention computation. Algorithm 1 demonstrates a parallel forward pass for one group of query heads, with multiple groups processed in parallel following the same approach. Particularly, we use softmax-off-by-one [34] to allow tokens in the current chunk to ignore any retrieved tokens. During the back-propagation process, we adopt a two-phased backward pass inspired by Dao [15]. The first stage accumulates gradients for $\mathbf{Q}$ and $\boldsymbol{w}$, followed by $\mathbf{K}$ and $\mathbf{V}$ in the second stage. The pseudo-code for this process is shown in Algorithm 2 and 3, in which $\mathbf{M}_{t,i} \in \{0,1\}^{l\times\lceil\frac{l}{S}\rceil}$ denotes whether the $t$-th token selects the $i$-th chunk, and $\mathbf{R}_{t,i} \in \mathbb{Z}^{l\times\lceil\frac{l}{S}\rceil}$ represents the index of the $i$-th chunk in $\mathcal{I}_t'$.

---

**Algorithm 1** FORWARD thread $t$

1: $\mathbf{O}_t' \leftarrow \mathbf{0}$      // *Initialize* $\mathbf{O}_t' \in \mathbb{R}^{h\times d_h}$
2: $\mathbf{Q}' \leftarrow$ load $\mathbf{Q}_t$      // *load* $\mathbf{Q}_t$ *to Static RAM (SRAM)*, $\mathbf{Q} \in \mathbb{R}^{l\times h\times d_h}, \mathbf{Q}' \in \mathbb{R}^{h\times d_h}$
3: **for** $1 \leq k \leq K$ **do**
4:     $i \leftarrow$ load $\mathcal{I}_{t,k}$, $w \leftarrow$ load $\boldsymbol{w}_{t,k}$      // $\mathcal{I} \in \mathbb{Z}^{l\times K}$, $\boldsymbol{w} \in \mathbb{R}^{l\times K}$
5:     $\mathbf{K}' \leftarrow$ load $\mathbf{K}_i$, $\mathbf{V}' \leftarrow$ load $\mathbf{V}_i$      // $\mathbf{K}, \mathbf{V} \in \mathbb{R}^{l\times h\times d_h} \mathbf{K}', \mathbf{V}' \in \mathbb{R}^{S\times d_h}$
6:     $\mathbf{O}' \leftarrow \mathrm{softmax}_1(\mathbf{Q}'\mathbf{K}'^\top)\mathbf{V}'$      // *Inter-chunk token-level attention, no online softmax required.*
7:     $\mathbf{O}_t' \leftarrow \mathbf{O}_t' + w\mathbf{O}'$      // *Chunk-level attention via weighted sum.*
8: **end for**
9: $\mathbf{O}_t' \rightarrow$ write to $\mathbf{O}_t$      // *Write to High Bandwidth Memory (HBM) from Static RAM (SRAM).*

---

**Algorithm 2** BACKWARD-$\mathbf{Q}, \boldsymbol{w}$ thread $t$

$\nabla\mathbf{Q}' \leftarrow \mathbf{0}, \mathbf{Q}' \leftarrow$ load $\mathbf{Q}_t, \nabla\mathbf{O}' \leftarrow$ load $\nabla\mathbf{O}_t$
**for** $1 \leq k \leq K$ **do**
   $i \leftarrow$ load $\mathcal{I}_{t,k}$, $w \leftarrow$ load $\boldsymbol{w}_{t,k}$
   $\mathbf{K}', \mathbf{V}' \leftarrow$ load $\mathbf{K}_i, \mathbf{V}_i$   // $\mathbf{K}', \mathbf{V}' \in \mathbb{R}^{S\times d_h}$
   $\mathbf{P} \leftarrow \mathrm{softmax}_1(\mathbf{Q}'\mathbf{K}'^\top)$   // $\mathbf{P} \in \mathbb{R}^{h\times S}$
   $\mathbf{O}' \leftarrow \mathbf{P}\mathbf{V}'$   // $\mathbf{O}' \in \mathbb{R}^{h\times d_h}$
   $\mathbf{D}' \leftarrow \mathrm{rowsum}(\mathbf{O}' \circ \nabla\mathbf{O}')$ // *pointwise multiply*
   // $\mathbf{D}' \in \mathbb{R}^h$, $\nabla\mathbf{O}' \in \mathbb{R}^{h\times d_h}$
   $\mathbf{D}' \rightarrow$ write to $\mathbf{D}_{t,k}$   // $\mathbf{D} \in \mathbb{R}^{l\times K\times h}$
   $\nabla w' \leftarrow \mathrm{rowsum}(\mathbf{D}')$   // $\nabla w' \in \mathbb{R}$
   $\nabla w' \rightarrow$ write to $\nabla\boldsymbol{w}_{t,k}$   // $\nabla\boldsymbol{w} \in \mathbb{R}^{l\times K}$
   $\nabla\mathbf{P} \leftarrow \nabla\mathbf{O}'\mathbf{V}'^\top$   // $\nabla\mathbf{P} \in \mathbb{R}^{h\times S}$
   $\nabla\mathbf{S} \leftarrow w\mathbf{P} \circ (\nabla\mathbf{P} - \mathbf{D}')$   // $\nabla\mathbf{S} \in \mathbb{R}^{h\times S}$
   $\nabla\mathbf{Q}_t \leftarrow \nabla\mathbf{Q}_t + \nabla\mathbf{S}\mathbf{K}'$   // $\nabla\mathbf{Q}_t \in \mathbb{R}^{h\times d_h}$
**end for**
$\nabla\mathbf{Q}' \rightarrow$ write to $\nabla\mathbf{Q}_t$

---

**Algorithm 3** BACKWARD-$\mathbf{K},\mathbf{V}$ thread $i$

$\mathbf{K}', \mathbf{V}' \leftarrow$ load $\mathbf{K}_i, \mathbf{V}_i$   // $\mathbf{K}', \mathbf{V}' \in \mathbb{R}^{S\times d_h}$
$\nabla\mathbf{K}', \nabla\mathbf{V}' \leftarrow \mathbf{0}$   // $\nabla\mathbf{K}', \nabla\mathbf{V}' \in \mathbb{R}^{S\times d_h}$
**for** $1 \leq t \leq l$ **do**
   **if** $\mathbf{M}_{t,i}$ is true **then**
     $k \leftarrow$ load $\mathbf{R}_{t,i}$, $w \leftarrow$ load $\boldsymbol{w}_{t,i}$
     $\mathbf{Q}' \leftarrow$ load $\mathbf{Q}_t$   // $\mathbf{Q}' \in \mathbb{R}^{h\times d_h}$
     $\nabla\mathbf{O}' \leftarrow$ load $\nabla\mathbf{O}_t$   // $\nabla\mathbf{O}' \in \mathbb{R}^{h\times d_h}$
     $\mathbf{D}' \leftarrow$ load $\mathbf{D}_{t,k}$   // $\mathbf{D}' \in \mathbb{R}^g$
     $\mathbf{P} \leftarrow \mathrm{softmax}_1(\mathbf{Q}'\mathbf{K}'^\top)$   // $\mathbf{P} \in \mathbb{R}^{h\times S}$
     $\nabla\mathbf{V}' \leftarrow \nabla\mathbf{V}' + w\mathbf{P}^\top\nabla\mathbf{O}'$
     $\nabla\mathbf{P} \leftarrow \nabla\mathbf{O}'\mathbf{V}'^\top$   // $\nabla\mathbf{P} \in \mathbb{R}^{h\times S}$
     $\nabla\mathbf{S} \leftarrow w\mathbf{P} \circ (\nabla\mathbf{P} - \mathbf{D}')$   // $\nabla\mathbf{S} \in \mathbb{R}^{h\times S}$
     $\nabla\mathbf{K}' \leftarrow \nabla\mathbf{K}' + \nabla\mathbf{S}^\top\mathbf{Q}'$
   **end if**
   $\nabla\mathbf{K}', \nabla\mathbf{V}' \rightarrow$ write to $\nabla\mathbf{K}_i, \nabla\mathbf{V}_i$
**end for**

## 3.2 Training & Inference

**Training.** Many efforts have been made to address Mamba's length generalization issues. A simple yet effective solution is truncated backpropagation through time (BPTT) [13, 66], in which the first state of a sequence is initialized as the final state of its preceding sequence. We follow this approach in training RAMba and other baselines. However, even though RAMba demonstrates certain extrapolation capabilities, we still observe an increase in perplexity on longer contexts. We hypothesize that RNNs' memory state provides certain shortcuts [22] for long-range attention, degrading performance on contexts largely exceeding the pre-trained length. Thus we attempt to introduce an appropriate forgetting mechanism into the memory state to disrupt the shortcuts. A straightforward way is memory reset, where sequences are divided into equal segments, and the initial state of each segment is reset to zero. To align with BPTT, we set the initial state as the last hidden state of a random segment in the previous step. In other words, for RNNs, the previous segment is randomly replaced, while the attention mechanism retains access to the original context. Our experiments show that this method effectively improves length generalization for RAMba.

**Inference.** A key challenge in introducing attention mechanisms to RNNs is managing the memory footprint, as the KV cache scales linearly with sequence length. Prior works [27, 35] have demonstrated the feasibility of offloading the KV cache to CPU memory and selectively loading chunks during inference. For a prompt of length $L$, all chunk representations are offloaded to the CPU after prefilling, while only $\mathbf{K}^{slc} \in \mathbb{R}^{\lfloor \frac{L}{S} \rfloor \times d}$ is kept in GPU memory for chunk selection. During decoding, RAMba retrieves and loads $K$ chunks at each step for each group. Since the KV cache is shared across all HSA layers, the number of parameters exchanged between the CPU and GPU totals $g \times d_h \times K \times S$. Our efficiency analysis in § 4.4 demonstrates the overhead for memory exchange is fully acceptable. Since the memory footprint of $\mathbf{K}^{slc}$ still increases with the sequence length, to theoretically achieve constant memory, a straightforward way is to offload it to a FAISS [19] database. However, in practice, the memory footprint of $\mathbf{K}^{slc}$ is very limited, making such offloading unnecessary. Detailed analysis is elaborated in the experiments section.

# 4 Experiments

## 4.1 Setups

To ensure a fair comparison, we pre-train all 370M models from scratch with 4K context length to observe their performance and extrapolation capabilities across various tasks. For 2.7B models, the training details are presented in Appendix E.

**Baselines.** We adopt the Mamba-2 architecture as the backbone of the RNN model and YaRN [44] as the Transformer baseline. The parameter size of all models trained from scratch is 370M, with detailed parameters provided in Appendix B. We experiment with Mamba variants with different attention mechanisms, including sliding window attention, native sparse attention (NSA), and HSA. For sliding window attention, the window size is set to 512, incorporating two position encoding schemes: ALiBi [45] and RoPE, the latter following the settings in Samba [51]. We set the chunk size of HSA to 64 following NSA. To ensure that the field of view for sparse attention matches the sliding window size (64 * 8 = 512), we set the number of selected chunks to 8. For NSA, we use its efficient open-source implementation [*]. To isolate the effects of the sparse attention components, we disable the sliding window attention in NSA. The HSA incorporates a single-layered Transformer-based bi-directional encoder for chunk memory encoding, accounting for 5.4% of the total parameters, whose impact on fairness is minimal. HSA layers are inserted into the upper decoder every $G = 8$ Mamba layers, with other attention mechanisms like SWA and NSA following the same pattern. These settings remain consistent across all subsequent 370M models. Since the compressed attention in NSA functions similarly to Combiner [50], we do not conduct a separate comparison against Combiner. Some other related works [36] are not included in the experiments due to the lack of open-source implementations.

**Pre-training.** All models are pre-trained on the same 60-billion-token subset of the Pile dataset [20]. Detailed training hyper-parameters are provided in Appendix C.

---

[*]https://github.com/fla-org/native-sparse-attention

| Models(370M) | pg19 | arxiv | code | pg19 | arxiv | code | pg19 | arxiv | code |
|---|---|---|---|---|---|---|---|---|---|
| | | eval_len=4k | | | eval_len=16k | | | eval_len=64k | |
| Transformer$_{\text{full\_attn}}$ | 18.61 | 4.23 | 3.28 | 539.15 | 199.42 | 62.17 | >10$^4$ | >10$^4$ | 2865.51 |
| Mamba | 17.92 | 4.24 | 3.28 | 17.38 | 3.91 | 3.09 | 17.30 | 3.86 | **3.05** |
| w/ SWA$_{\text{ALiBi}}$ | 17.82 | 4.21 | 3.26 | 20.48 | 5.01 | 3.53 | 23.86 | 6.46 | 3.96 |
| w/ SWA$_{\text{rope}}$ | 17.82 | 4.21 | 3.26 | 17.50 | 4.03 | 3.19 | 17.80 | 4.25 | 3.35 |
| w/ NSA$_{\text{w/ m.r.}}$ | 17.87 | 4.20 | 3.25 | 17.31 | 3.87 | 3.06 | 17.31 | 3.87 | **3.05** |
| w/ NSA$_{\text{w/o m.r.}}$ | 17.74 | 4.18 | 3.24 | 17.56 | 4.29 | 3.26 | 17.62 | 4.35 | 3.28 |
| RAMba$_{\text{w/ m.r.}}$ | 17.82 | 4.15 | 3.23 | 17.15 | **3.73** | **3.04** | **17.01** | **3.65** | 3.07 |
| RAMba$_{\text{w/o m.r.}}$ | **17.63** | **4.13** | **3.21** | **17.11** | 3.81 | 3.08 | 17.11 | 3.87 | 3.21 |
| RAMba$_{\text{w/o m.r., w/o s.b.}}$ | 18.07 | 4.52 | 3.34 | 17.61 | 5.01 | 3.17 | 17.61 | 6.05 | 3.16 |

Table 1: Perplexity for long-range language modeling. We highlight the best results in **bold** and underline the second best. All models are pre-trained on 4K contexts.

**Ablation Studies.** We denote memory reset usage as **w/ m.r.** and its absence as **w/o m.r.**. Additionally, **w/o s.b.** refers to using softmax without positional encoding instead of stick-breaking weights. For an 8K token context, the memory is reset every 4K tokens, which aligns with the context length of other baselines. However, the chunk selection scope for sparse attention spans 8K tokens, which might be unfair to other baselines. To ensure fair comparisons, we apply the same settings to both HSA and NSA.

## 4.2 Long-range Langauge Modeling

**Datasets.** We evaluate long-range language modeling on PG19 [48], ArXiv-math [4], and Code [65].

**Results.** As shown in Table 1, RAMba performs better than the baselines across most datasets, on both in-domain (4K) and out-of-domain (16K & 64K) lengths. In addition, we have two findings. **First**, Mamba's perplexity decreases as the context length increases from 4K to 64K, whereas Mamba with attentions shows higher perplexity at 64K compared to 16K when not trained with memory reset. This suggests that Mamba's memory state may influence the generalization ability of attention mechanisms. **Secondly**, when trained with memory reset, Mamba with NSA or HSA all exhibit stronger length extrapolation capabilities but show a decline in in-domain performance. This result validates the effectiveness of memory reset, which essentially constrains the model to rely entirely on the content of the text for retrieval, thereby preventing the model from learning shortcuts. Since the model becomes adapted to retrieving information from longer contexts, it is reasonable that its performance declines when handling shorter contexts.

## 4.3 Downstream Tasks

**Tasks.** We evaluate various models' long-context modeling abilities on classic tasks like passkey retrieval [35] and the LongBench V2 dataset [7]. To increase task difficulty, we replace numbers in passkey retrieval with random token sequences. Since passkey retrieval is relatively simple, we further fine-tuned the models using synthetic data following RULER [25]. We use a context length of 4K for fine-tuning with a total training step size equivalent to 5% of the pre-training stage. Evaluations were conducted across different lengths on four RULER tasks: Single NIAH (S-N), Multi-queries NIAH (MQ-N), Variable Tracking (VT), and Frequent Words Extraction (FWE). To align with passkey retrieval, keys in Single NIAH were also replaced with random token sequences. We adopt a Cloze format for LongBench evaluation, following Waleffe et al. [62], to address the instruction-following challenges of small models. Since LongBench V2 is a zero-shot benchmark and thus small models may exhibit randomness, we additionally evaluate on fine-tunable datasets, including summarization tasks like XSUM [39] and CNN [38], and QA tasks like SQuaD [49], HotpotQA [68], and QuALITY [40]. NSA's implementation currently does not support generation, so its results on generative tasks are not reported.

**Results.** **First**, Figure 3 shows that RAMba achieves perfect accuracy in the passkey retrieval task with a 64M context, even without memory reset, while most baselines drop to nearly zero around 64K length. However, removing the chunk encoder causes RAMba's accuracy to decrease as context length grows, highlighting the encoder's importance for HSA and length generalization. One possible reason is that the hidden states are primarily optimized for predicting the next token, thus

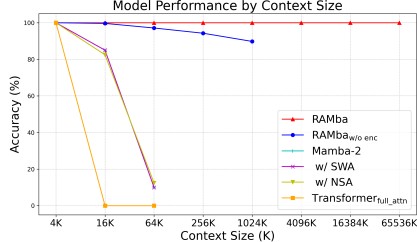

Figure 3: Passkey retrieval results.

| Models(370M) | Overall | Easy | Hard | Short | Medium | Long |
|---|---|---|---|---|---|---|
| Transformer$_{full\_attn}$ | 24.8 | 23.4 | 25.7 | **28.5** | 19.6 | **29.5** |
| Mamba | 22.4 | 16.4 | **26.1** | 19.6 | 22.7 | 26.3 |
| w/ SWA$_{ALiBi}$ | 23.3 | 26.9 | 21.0 | 19.6 | 26.8 | 22.1 |
| w/ SWA$_{rope}$ | 24.2 | 25.7 | 23.2 | 20.3 | 26.3 | 26.3 |
| w/ NSA$_{w/ m.r.}$ | 23.5 | 25.7 | 22.1 | 21.5 | 25.8 | 22.1 |
| w/ NSA$_{w/o m.r.}$ | 22.1 | 26.3 | 19.6 | 17.1 | 26.3 | 22.1 |
| RAMba$_{w/ m.r.}$ | **25.7** | **28.7** | 23.9 | 20.9 | **27.8** | **29.5** |
| RAMba$_{w/o m.r.}$ | 23.5 | 26.3 | 21.7 | 20.9 | 23.7 | 27.4 |

Table 2: LongBench V2 results.

| Models (370M) | S-N | MQ-N | VT | FWE | S-N | MQ-N | VT | FWE | S-N | MQ-N | VT | FWE | S-N | FWE |
| --- | --- | --- | --- | --- | --- | --- | --- | --- | --- | --- | --- | --- | --- | --- |
| | | ctx-len=4K | | | | ctx-len=64K | | | | ctx-len=256K | | | ctx-len=1M | |
| Transformer$_{full\_attn}$ | 95.08 | 88.59 | **97.12** | 60.02 | 0.00 | 0.00 | 0.00 | 0.00 | 0.00 | 0.00 | 0.00 | 0.00 | 0.00 | 0.00 |
| Mamba-2 | **96.66** | 1.86 | 61.78 | 64.75 | 10.45 | 0.00 | 8.96 | 37.31 | 0.00 | 0.00 | 6.25 | **43.75** | 0.00 | **40.00** |
| w/ SWA$_{rope}$ | 91.84 | 6.96 | 37.20 | 72.08 | 28.36 | 0.00 | 17.91 | 0.00 | 6.25 | 0.00 | 0.00 | 0.00 | — | — |
| w/ NSA (w/o m.r.) | 92.58 | 59.09 | 57.70 | 66.98 | 13.43 | 0.00 | 0.00 | 2.99 | 6.25 | 0.00 | 0.00 | 6.25 | — | — |
| w/ NSA (w/ m.r.) | 84.79 | 4.08 | 31.63 | 49.54 | 4.48 | 0.00 | 5.97 | 26.87 | 0.00 | 0.00 | 0.00 | 12.50 | — | — |
| RAMba$_{w/ m.r.}$ | 91.74 | 80.61 | 95.18 | 49.81 | **85.07** | **55.22** | 55.22 | **53.73** | 62.50 | 62.50 | 37.50 | 37.50 | 24.00 | 20.00 |
| RAMba$_{w/o m.r.}$ | 92.76 | 87.10 | 96.66 | **76.81** | 55.22 | 29.85 | **68.66** | 8.96 | 18.75 | 12.50 | 31.25 | 0.00 | — | — |
| RAMba$_{w/o enc}$ | 92.67 | 81.08 | 57.79 | 75.97 | 41.79 | 23.88 | 14.92 | 26.87 | 6.25 | 12.50 | 0.00 | 6.25 | — | — |

Table 3: Results for selected sub-tasks in RULER. All models are pre-trained on 4K contexts.

requiring an adapter to extract information representing the current context. **Second**, LongBench's evaluation results in Table 2 are generally consistent with those of language modeling. However, since LongBench is a zero-shot benchmark, most baselines performed below the random guess rate of 25%, with the results across different sub-tasks show high variability. Thus, we believe the following SFT tasks can provide deeper insights into architectural capabilities for small models.

Table 3 presents the evaluation results of various models fine-tuned on RULER synthetic data across different context lengths. We have four interesting findings. **First, HSA demonstrates more precise chunk selection.** HSA performs comparably to full-attention on retrieval-related tasks (S-N, MQ-N, VT) over in-domain context length, while NSA lags behind both, supporting our argument that estimating chunk importance using token-to-token attention scores is inaccurate. Meanwhile, HSA achieves strong performance across retrieval tasks despite selecting chunks only once, validating the effectiveness of the hierarchical attention mechanism in learning token-to-chunk relevance. **Second, Mamba excels in sequential statistical tasks.** While Mamba-based models underperform in retrieval tasks, they outperform in Frequent Word Extraction (FWE), successfully extrapolating up to $256\times$ the pre-training length. This advantage likely stems from Mamba's continuous memory flow mechanism, which is also partially inherited by RAMba. **Third, memory reset helps length generalization.** Models with memory reset consistently outperform those without in length extrapolation. Though performance drops significantly beyond 1M context length on more challenging retrieval tasks, it still achieves $256\times$ extrapolation. **Fourth, templates can make a significant difference.** Single-NIAH extrapolates only to 4M, while the similar task of passkey retrieval extends to 64M, with the sole difference lying in the template of passkey retrieval being simpler. We elaborate the templates of these two tasks in Appendix D. It suggests that a simpler pattern may facilitate learning more generalizable patterns. Still, precise chunk selection for extremely long contexts remains an open challenge for future work.

| Models (370M) | XSUM$_{R-1/R-2/R-L}$ avg-len=498 | CNN$_{R-1/R-2/R-L}$ avg-len=883 | SQuaD$_{EM/F1}$ avg-len=174 | HotpotQA$_{EM/F1}$ avg-len=1428 | QuALITY$_{Acc}$ avg-len=7745 | AVG. |
|---|---|---|---|---|---|---|
| Mamba-2 | 30.40/11.89/24.53 | 37.97/17.54/35.84 | 41.33/52.03 | 18.70/26.20 | 33.32 | 31.03 |
| w/ SWA$_{ALiBi}$ | 30.78/12.11/24.79 | 38.96/17.84/36.73 | 50.54/61.23 | 19.96/27.58 | 29.48 | 32.57 |
| w/ SWA$_{rope}$ | **30.88/12.33/24.99** | 38.78/17.67/36.57 | 51.46/61.91 | 19.89/28.00 | 32.89 | 32.61 |
| w/ NSA | — | — | — | — | 31.40 | — |
| Transformer$_{full\_attn}$ | 30.10/11.63/24.23 | 38.25/17.89/36.14 | 45.50/56.13 | 21.90/29.49 | 33.46 | 32.82 |
| RAMba | 30.81/12.25/24.80 | **39.11/18.04/36.85** | 48.24/59.17 | **22.30/30.53** | **34.13** | **33.64** |

Table 4: Downstream task evaluations.

Table 4 shows the results of the model after fine-tuning on summarization and generation tasks. RAMba still outperforms in the vast majority of tasks, but two observations are worth noting. **First**, Mamba with SWA demonstrates advantages over other models on the SQuaD task. One potential reason is that the SQuaD dataset contains shorter texts, even less than the sliding window, making

SWA able to randomly access all contexts. This phenomenon shows that integrating random access can improve RNN performance even for short contexts. **Second**, despite lagging behind Mamba in perplexity (PPL), the Transformer demonstrates strong downstream performance, ranking just behind RAMba. This further validates the importance of random access capability for downstream tasks.

| Models | #params. | LongBench | | | | | | RULER$_{\text{S-N/MQ-N/VT/FWE}}$ | | |
| | | Overall | Easy | Hard | Short | Medium | Long | ctx-len=4K | ctx-len=64K | ctx-len=1M |
|---|---|---|---|---|---|---|---|---|---|---|
| Mamba-2 | 2.7B | 26.8 | 23.4 | 29.0 | 29.7 | 27.3 | 21.1 | 89.24/26.62/24.68/**77.09** | 0.00/0.00/0.00/0.00 | 0.00/0.00/0.00/0.00 |
| RAMba$_{\text{w/ m.r.}}$ | 2.7B + 110M | **27.3** | 25.1 | 28.6 | 30.4 | 27.3 | 22.1 | **91.00/86.36/96.47**/42.95 | 83.58/76.12/100.00/28.36 | 25.00/13.33/50.00/37.14 |

Table 5: Results of 2.7B models trained on 4K contexts.

Table 5 shows the results of the 2.7B models. These results show that even after scaling up, RAMba still maintains a significant lead. Details about training can be found in Appendix E.

## 4.4 Efficiency Analysis

**Experimental Setup.** To evaluate the scalability and efficiency of HSA, we measure the runtime and memory footprint of FlashAttention-2, NSA, and HSA operators across different sequence lengths, the throughput of various models at different scales, and the per-token time consumption during inference. Runtime measurements were conducted with three attention layers only, excluding additional components such as MLPs or Mamba. In HSA, a single chunk selection is shared across all three layers, while SWA remains disabled in NSA. The memory footprint is reported as a ratio of the memory occupied by Mamba 370M. We use **w/ offloading** and **w/o offloading** to denote whether memory offloading is enabled. To further evaluate the benefits of shared chunk selection, we introduce an ablation group (**w/o sharing**), where chunk selection and CPU-GPU memory exchanges happen in each HSA layer. When measuring training throughput, we enable FSDP [75] and gradient checkpointing [12], running models on 16 × Physics Processing Units (PPUs), each with approximately half the computational power of an A100 GPU.

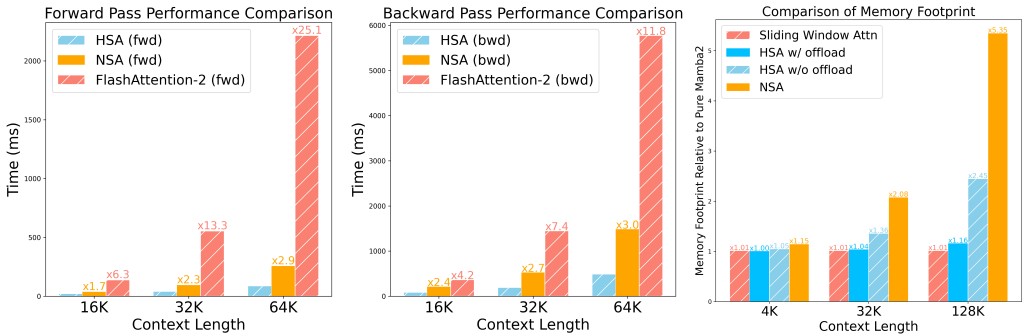

Figure 4: Comparison of attention computation time: 3 attention layers per group. (The lower the better)

| Models | Parameter Size | | | |
| | 370M↑ | 780M↑ | 1.4B↑ | 3B↑ |
|---|---|---|---|---|
| Mamba | 16.44 | 9.71 | 6.53 | 3.46 |
| w/ SWA | 15.64 | 9.24 | 6.27 | 3.31 |
| w/ full_attn | 10.00 | 5.73 | 4.31 | 2.07 |
| w/ NSA | 14.35 | 8.70 | 5.46 | 3.18 |
| RAMba | 14.80 | 8.76 | 5.70 | 3.18 |

Table 6: Training throughput ($10^3$ tokens/s) with context length=32K.

| Models | Prompt-Length | | |
| | 4K↓ | 16K↓ | 64K↓ |
|---|---|---|---|
| Transformer$_{\text{full\_attn}}$ | 2.26 | 8.90 | 32.12 |
| Mamba-2 | 2.92 | 2.82 | 2.84 |
| RAMba$_{\text{w/o offloading}}$ | 3.14 | 3.05 | 2.76 |
| RAMba$_{\text{w/ offloading}}$ | 3.97 | 4.19 | 4.02 |
| RAMba$_{\text{w/ offloading, w/o sharing}}$ | 5.98 | 5.87 | 5.95 |

Table 7: Inference time cost (seconds, prefilling time excluded) for generating 100 tokens (batch-size=16)

**Results.** Figure 4 compares the time consumption and memory footprint of three operators across different context lengths. Both NSA and HSA, as sparse attention mechanisms, significantly outperform Flash-Attention in terms of speed. HSA is faster than NSA because it performs chunk selection—the only operation with quadratic complexity—only once, and shares it across all HSA layers. In contrast, NSA involves computations with quadratic complexity like compressed token attention and chunk selection in every layer, which increases its time consumption. For memory footprint, enabling memory offloading drastically reduces GPU memory usage and slows its growth with

increasing context length. Despite introducing CPU-GPU memory exchange, the impact on inference speed is limited, as reported in Table 7. Table 6 shows that RAMba achieves 90% of Mamba's training throughput. Although HSA is faster than NSA, the additional encoder sometimes offsets this advantage at certain scales. Nonetheless, the results clearly highlight HSA's high efficiency, excellent scalability during training, and near-constant memory usage during inference.

## 5   Conclusion

In this work, we present Hierarchical Sparse Attention (HSA) and build on it to propose RAMba, which integrates an RNN backbone, length-generalizable sparse attention, and a simple forgetting mechanism. This architecture strikes a strong balance between performance, efficiency, long-range access, and length generalization, offering a foundation for language models with permanent memory.

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

## Limitations

Although the method proposed in this work is theoretically applicable to all RNNs, the experimental section of this work mainly focuses on Mamba.

Considering computational resources, this work does not discuss the performance of models larger than 3B parameters.

## A  How HSA achieves accurate chunk retrieval

The core principle of previous sparse attentions' chunk selection lies in approximating the token-to-chunk relevance using unnormalized attention scores. In self-attention, the relevance of token $j$ to token $i$ is defined as:

$$p_{i,j} = \frac{e^{\mathbf{logits}_{i,j}}}{\mathbf{Z}_i} , \mathbf{Z}_i = \sum_{j<i} e^{\mathbf{logits}_{i,j}}, \tag{1}$$

where $\mathbf{logits}_{i,j} = \mathbf{q}_i^\top \mathbf{k}_j$ is the dot product between the query $\mathbf{q}_i$ of token i and the key $\mathbf{k}_j$ of token j.

The relevance of chunk $c$ to token $i$ is ideally the sum of the relevance of all tokens within that chunk:

$$r_{i,c} = \sum_{j\in\mathcal{C}} p_{i,j} = \frac{1}{\mathbf{Z}_i} \sum_{j\in\mathcal{C}} e^{\mathbf{logits}_{i,j}} , \tag{2}$$

where $\mathcal{C}$ denotes tokens in chunk $c$. However, calculating $r_{i,c}$ requires computing $\mathbf{Z}_i$ (the full softmax normalization across all tokens), which would necessitate full computation and thus undermine the computational efficiency.

To ensure efficiency, they instead approximate chunk relevance using the mean-pooled representation of keys:

$$r'_{i,c} = \mathbf{q}_i^\top \mathbf{K}_c = \mathbf{q}_i^\top \frac{1}{S} \sum_{j\in\mathcal{C}} \mathbf{k}_j = \frac{1}{S} \sum_{j\in\mathcal{C}} \mathbf{q}_i^\top \mathbf{k}_j = \frac{1}{S} \sum_{j\in\mathcal{C}} \mathbf{logits}_{i,j} , \tag{3}$$

where $\mathbf{K}_c$ represents the mean-pooling of key representations within the $c$-th chunk. While this approximation bypasses the need for normalized softmax scores across all tokens, it introduces a discrepancy between $r'_{i,c}$ (the approximation) and $r_{i,c}$ (the ideal one).

Let's consider the following example shown in Figure 5, assuming every 2 tokens form a chunk. If

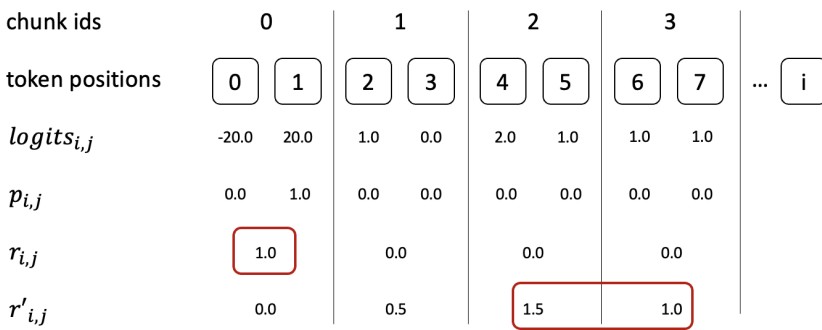

Figure 5: How unnormalized scores mislead the chunk selection.

only the top-2 chunks can be selected, they would choose chunks 2 and 3 according to $r'_{i,c}$, thus missing the chunk 0 with the highest sum of attention weights.

Assuming they select the top-2 chunks for sparse attention, the subsequent attention process is shown in Figure 6. Throughout this process, $r'_{i,c}$ only participates in chunk selection but not in the forward computation, nor does it receive gradients. The inaccurate chunk selection issue stems from using unnormalized attention logits to estimate chunk importance, with $r'_{i,c}$ not learnable, making the inaccuracy of chunk selection unavoidable.

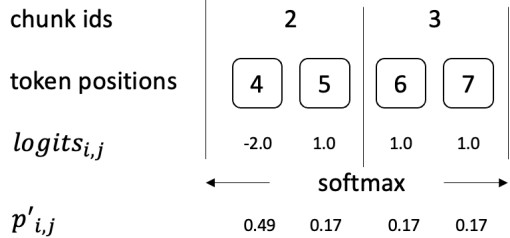

Figure 6: Applying attention over the concatenation of selected chunks.

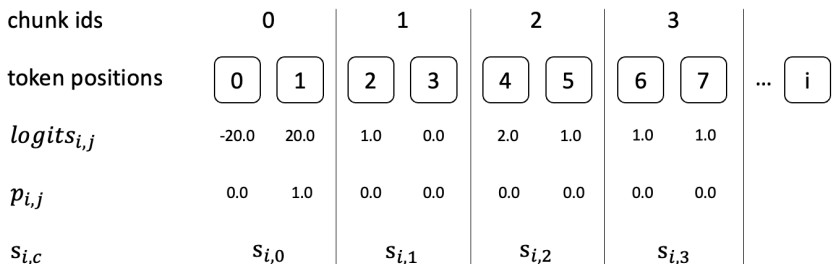

Figure 7: The chunk selection stage of HSA

For the same example, HSA works as shown in Figure 7 where $s_{i,c}$ represents learnable token-to-chunk relevance scores. HSA selects chunks 2 and 3 according to $s_{i,c}$, and applies the hierarchical attention over selected chunks as shown in Figure 8. In this process, $w_{i,c}$, derived from $s_{i,c}$, par-

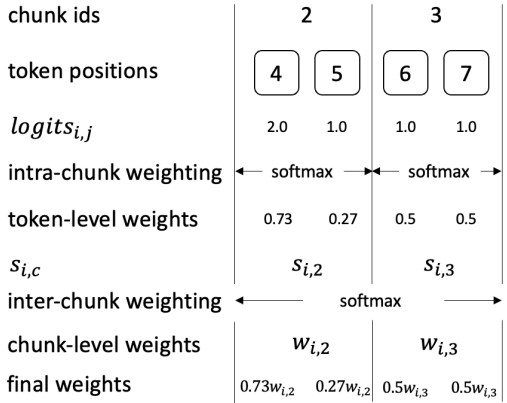

Figure 8: Applying hierarchical attention over selected chunks.

ticipates in the final attention weight computation and the whole forward pass. This allows it to receive gradients and allocate higher weights to chunks with more important tokens. Even if random initialization initially misses the most relevant chunk 0, continuous training enables the model to gradually learn to select the most relevant chunks.

## B  Hyper-parameters

| Architecture | Transformer | Mamba | Mamba+NSA | Mamba+SWA | RAMba |
|---|---|---|---|---|---|
| Total Params (M) | 372 | 368 | 375 | 375 | 385 |
| Hidden size, $d$ | 1024 | 1024 | 1024 | 1024 | 1024 |
| Mamba Layers | - | 48 | 48 | 48 | 48 |
| Attention Layers | 16 | - | 3×NSA | 3×SWA | 3×HSA |
| Other Layers | 16×MLP | - | - | - | 1×Chunk Selection, 1×Encoder |
| MLP hidden size | 5504 | - | - | - | 1344 |
| Query heads | 32 | - | 16 | - | 16 |
| KV heads | 16 | - | 1 | - | 1 |
| Vocab size | 50280 | 50280 | 50280 | 50280 | 50280 |

Table 8: Hyper-parameters of 370M models.

## C  Training hyper-parameters

All 370M models used the **AdamW** optimizer with

- linear learning rate warmup with warmup ratio 0.02, cosine decay to $4e-5$.
- peak learning rate $2e-3$.
- total tokens $60B$, batch size $1M$ tokens.
- gradient clip value 1.0
- no dropout
- no linear bias terms
- weight decay $1e-3$
- AdamW hyperparameter $\beta = (.9, .95)$ (the GPT3 value)

All models are pre-trained on 16 PPUs, with each taking approximately 60 hours.

## D  Templates for passkey retrieval and Single-NIAH tasks

The passkey retrieval template is structured as follows: "(essays) The pass key is <PASS KEY>. (essays) What is the passkey? The passkey is". The single NIAH template follows this format: "(essays)... One of the special magic numbers for long-context is: <PASS KEY>. (essays)... What is the special magic number for long-context mentioned in the provided text? Answer:"

## E  RAMba 2.7B

We follow the hyperparameters of Mamba-2 2.7B, where the embedding dimension is 2560, and the total number of layers is 64. We use a two-layer Transformer-based encoder with a hidden size of 2560 and an intermediate dimension of 3392. HSA is inserted into Mamba starting from the 32nd layer, with one HSA layer inserted every 8 layers. In HSA, the group size $g$ is 5, with a query head size $h$ of 16 for each group. The total additional parameters amount to approximately 110M.

### E.1  Post-training

Due to the length generalization issues inherent in the Mamba-2 2.7B, we post-train it to stabilize its perplexity on longer contexts. Specifically, we utilized BPTT for post-tuning the base model. We trained the model on sequences of 32K tokens with a batch size of 16 for 3K steps, totaling 1.5B tokens. This stage takes 5 hours on 32 PPUs.

Then we follow CEPE [69] by freezing most parameters of the Mamba backbone and tuning the parameters of the additional HSA modules for post-training. This process primarily involves the following two stages:

**Warmup.** We employ a warmup initialization method by simply training the model to copy the first half of the sentence. Specifically, we append an identical copy of each sentence to itself and train the model to locate the distant context and replicate it. At this stage, all parameters of Mamba-2 are frozen, with only the HSA-related parameters remaining tunable. We train the model with a 32K context length, batch size of 16, for 16K steps, with a peak learning rate of $2 \times 10^{-5}$, totaling 8B tokens. This stage takes around 24 hours on 32 PPUs.

**Post-Training.** We use LoRA [26] to fine-tune 5% of the parameters in the Mamba-2 module, while keeping all parameters in the HSA module fully trainable. The model is trained with a context length of 32K, a batch size of 16, for 32K steps, using a peak learning rate of $2 \times 10^{-5}$, totaling 16B tokens. This stage takes around 48 hours on 32 PPUs.

This model is used for the LongBench evaluation in Table 5.

## E.2 RULER finetuning

Since the RULER tasks like NIAH and FWE do not have high requirements for intrinsic knowledge of LLMs, we opt to train RAMba from scratch and fine-tune it on RULER's synthetic dataset. This approach aims to evaluate whether RAMba trained from scratch can stably converge and demonstrate long-range retrieval capabilities. We conduct pre-training on 60B tokens, which amounts to one-tenth of the Mamba-2 2.7B model, followed by fine-tuning on 1B synthetic data, which takes around 200 hours on 32 PPUs. We also fine-tuned the Mamba-2 2.7B model on the same synthetic dataset for comparison.

