# OpenReview forum: "Hardware-aligned Hierarchical Sparse Attention for Efficient Long-term Memory Access"
_NeurIPS.cc/2025/Conference — NeurIPS 2025 poster_

### Official Review · Reviewer_XZCu · 2025-06-29

**Clarity:** 3
**Significance:** 3
**Originality:** 3
**Rating:** 5
**Confidence:** 3

**Summary:**

This paper claims that current RNNs fail to randomly access historical context and questions whether the efficiency advantages still hold when attention mechanisms are integrated into RNNs. To this end, the authors propose Hierarchical Sparse Attention and combine it with Mamba2, leading to a hybrid RAMba architecture. HSA consists of two stages, including a bidirectional encoder within each chunk and a weighted summation for the chunks selected by top-k mechanisms. Moreover, they design a hardware-aligned Triton kernel to efficiently implement HSA. RAMba achieves remarkable performance in length generalization, performance, and efficiency, pushing ahead with the boundaries of long-context modeling.

**Questions:**

1. Why does HSA adopt the stick-breaking attention mechanism for constructing chunk weights, rather than other forms of attention with positional encoding? I would like to see a theoretical justification or supporting ablation studies. (Table 1 presents RAMba w/o m.r. w/o s.b., which uses softmax without positional encoding, but it does not include ablation related to positional encoding)

2. Line 174: Does the equation for the HSA layer in this line omit the Mamba layers? I ask because the formula shows H_t^(l+1)=H_t^l+O_t^l, but in the architecture, there are components other than the HSA layer between H_t^(l+1) and H_t^l, including Mamba layers.

3. Line 273: Why does NSA's implementation currently not support generation?

**Ethical Concerns:**

["NO or VERY MINOR ethics concerns only"]

**Final Justification:**

The authors have addressed the concerns I raised. Overall, to enhance the ability of RNNs to accurately retrieve historical information, the authors introduce a new sparse attention mechanism that facilitates precise chunk selection. Through the design of a hybrid architecture, they demonstrate that the proposed method achieves a balance among extrapolation ability, performance, and efficiency. The remaining issues for the authors are limited to refinements in methodology and some wording.

**Limitations:**

I agree that experiments with RNNs besides Mamba can be explored further.

**Quality:**

3

**Strengths And Weaknesses:**

Strengths

1. A sparse attention design with a two-stage hierarchical structure. Chunk-level information is captured in the first stage and then fused in the second stage, which helps the length generalization. The working memory and long-term memory from a psychological perspective are interesting.

2. A hardware-aligned algorithm to achieve efficient parallel computation.

3. Extensive experiments on long-range language modeling, Passkey retrieval, LongBench, RULER, and fine-tunable datasets. Results of 2.7B models on LongBench and RULER validate the scalability. The result that the running time and memory consumption outperform NSA is encouraging, thanks to the shared chunk memory design. Additionally, the ablation studies regarding memory reset and sticking weights are thorough.

weakness

1. The motivation and the solution of the paper are not tightly aligned. My main concern is that the authors use "Random Long-context Access for Mamba" as the central motivation, while the main focus of the paper is actually on proposing the HSA method. In fact, the authors do not design any sparse mechanism specifically for the Mamba block itself, nor do they provide an in-depth discussion on the relationship between HSA and Mamba. Instead, HSA is simply applied within a hybrid architecture. As a result, this narrative may dilute the core contribution of the work. Moreover, I believe that using "random access" to describe chunk selection may be misleading, although the phrase contributes to the formation of the acronym "RAMba".

2. Line 29-30: Is there any literature supporting the claim that augmenting RNNs with attention mechanisms leads to poor length extrapolation? The authors appear to assume that hybrid models extrapolate worse than pure RNNs, but this assumption is questionable. To the best of my knowledge, hybrid architectures do not necessarily hinder length extrapolation [1].

3. Lines 139 and 157: Does the introduction of the transformer-based bi-directional encoder and the mean-pooled representation compromise the model's causality? MoBA [2] points out: "The routing to the current block could also violate causality, since mean pooling across the entire block can inadvertently include information from future tokens." To address this, they ensure that each token is routed to its respective current block and apply a causal mask during the current block attention. I would be glad to see the authors' perspective on this issue.

4. The results in Figure 3 are excellent. However, I have one question: What mechanism enables RAMba to achieve better extrapolation performance than both Mamba-2 and NSA? Even when the encoder is removed, the results remain strong. If the authors could provide a reasonable explanation for this significant performance gain, it might offer useful insights for future architecture design.

5. Can HSA operate independently of the Mamba blocks? If a model were designed using only HSA, would it still be faster than NSA?

[1] MiniMax-01: Scaling Foundation Models with Lightning Attention.
[2] MoBA: Mixture of Block Attention

---

> ### Author Rebuttal · Authors · 2025-07-28
>
> We sincerely appreciate your insightful feedback.
>
> W1. **Regarding Motivation, the authors do not design any sparse mechanism specifically for the Mamba block itself**
>
> We believe that "Enabling Random Long-Context Access for Mamba" is a goal, and the sparse mechanism for Mamba block could be one way to achieve it, but maybe not the only way. This paper, "via Hierarchical Sparse Attention," presents another approach to achieving this goal through a hybrid architecture.
>
> As mentioned in our abstract and introduction, there is an information bottleneck in RNNs due to compressing variable-length sequences into fixed-dimensional representations, which inevitably leads to information loss. In contrast, the kv cache in attention grows with sequence length, which has no information bottleneck.
>
> Thus, the current mainstream approach to enhancing Mamba's long-range random access ability is integrating Mamba with full attention [1]. As stated in lines 32-33 of the introduction, this leads to a trilemma. Consequently, our motivation lies in replacing full attention with sparse attention, as reflected in *"Enabling Random Long-Context Access for Mamba via HSA."*
>
> **Regarding Random Access**
>
> "Random access" refers to the ability to access any arbitrary past token, not just selecting chunks. For example, in an open-book exam, you locate a page (chunk) and then pinpoint specific details (tokens). Similarly, HSA enables access to any past token by first identifying the chunk and then the token-level details.
>
> W4. **What mechanism enables RAMba to achieve better extrapolation performance than both Mamba-2 and NSA?**
>
> We believe this is an interesting and important topic, so let us discuss it in advance.
>
> First of all, the underperformance of Mamba-2 has been discussed in many studies [5]. The main reason is the information bottleneck issue, as we mentioned above.
>
> To know why HSA achieves better extrapolation performance, let's take a close look at how NAS and HSA work in the following example.
> Assuming every 2 tokens form a chunk, separated by |:
> ```
> chunk ids             |       0      |      1    |      2    |     3     | ...
> token positions       |   0   |  1   |  2  |  3  |  4  |  5  |  6  |  7  | ... , i <- the currrent token
> attn logits[i, *]     | -20.0 | 20.0 | 1.0 | 0.0 | 2.0 | 1.0 | 1.0 | 1.0 |
> attn weights (softmax)|   0.0 |  1.0 | 0.0 | 0.0 | 0.0 | 0.0 | 0.0 | 0.0 |  // O(L$^2$), L: sequence length
> chunk attn logits sum |      0.0     |    0.5    |    1.5    |    1.0    |  // O(L*(L/S), S: chunk size
> ```
> NSA selects chunks according to "chunk attn logits sum" for efficiency.
> If only the top-2 chunks can be selected, NSA would choose chunks 2 and 3, thus missing the chunk 0 with the highest sum of attention weights, **leading to inaccurate chunk selection**.
>
> Assuming NSA selects the top-2 chunks for sparse attention, the following process would be:
> ```
> chunk ids         |      2      |      3      |
> token positions   |   4  |   5  |   6  |   7  |
> attn logits       |  2.0 |  1.0 |  1.0 |  1.0 |
>                   |<-----     softmax   ----->|
> attn weights      | 0.49 | 0.17 | 0.17 | 0.17 |
> ```
>
> Throughout this process,  chunk selection can not be explicitly learned.
>
> For the same example, HSA works as follows:
> ```
> chunk ids          |       0      |      1     |      2    |     3     | ...
> token positions    |   0   |   1  |  2   |  3  |  4  |  5  |  6  |  7  | ...
> attn logits        | -20.0 | 20.0 |  1.0 | 0.0 | 1.0 | 1.0 | 1.0 | 1.0 | ...
> s[i,c]             |    s [i,0]   |  s [i,1]   |  s [i,2]  |  s [i,3]  | ...
> ```
> Where s [i,c] represents learnable token-to-chunk relevance scores as described in Line 155 of the paper. For example, HSA may also select chunks 2 and 3 when parameters are randomly initialized, we have:
> ```
> chunk ids            |           2           |          3          |
> token positions      |     4     |     5     |     6    |     7    |
> attn logits          |   2.0     |    1.0    |    1.0   |    1.0   |
> intra-chunk weighting|<--     softmax      ->|<--    softmax     ->|
> token-level weights  |   0.73    |    0.27   |    0.5   |    0.5   |
> s[i,c]               |        s[i,2]         |        s[i,3]       |
> inter-chunk weighting|<---            stick-breaking           --->|
> chunk-level weights  |        w[i,2]         |        w[i,3]       |   // s.t. w [i,2} + w [i,3] <= 1
> final weights        |0.73×w[i,2]|0.27×w[i,2]|0.5×w[i,3]|0.5×w[i,3]|
> ```
>
> In this process, w[i,c], derived from s[i,c], participates in the final attention weight computation and the whole forward pass. This allows it to receive gradients and allocate higher weights to chunks with more important tokens. Even if random initialization initially misses the most relevant chunk 0, continuous training enables the model to gradually learn to select the most relevant chunks.
>
> Such a learnable chunk selection mechanism is essentially a kind of token-to-chunk retrieval, which may lead to better extrapolation performance
>
> W2. **Is there any literature supporting the claim that augmenting RNNs with attention mechanisms leads to poor length extrapolation?**
>
> Yes.
> Specifically, lines 29-30 refer to hybrid architectures like Mamba with full attention. Mamba with sliding window attention is not included in this discussion. The issue stems from the poor length extrapolation ability of full-attention, which has never been fundamentally solved.
>
> For instance, Figure 1 of YaRN [4] shows how perplexity (ppl) increases when full-attention-based models with RoPE extrapolate to unseen context lengths.
>
> Table 2 in [7] also reveals that, even with temperature scaling and discard positional encodings, perplexity still dramatically increases beyond a certain length, indicating poor extrapolation.
>
> The existing full-attention-based models with 1 million context lengths primarily rely on techniques like ring-attention to brute-force scale the pretraining length, which requires huge computational resources.
>
> As minimax[6] mentioned in their abstract, "The context window of MiniMax-Text-01 can reach up to 1 million tokens **during training** and extrapolate to 4 million tokens during inference at an affordable cost. ” They also acknowledge the limitation of only being able to extrapolate up to 4 times the training length. The 1 million pre-training context window is also achieved via ring-attention.
>
> Our work achieves extrapolation from 4K to 64M (16000×) in passkey retrieval and maintains high performance in both training and inference.
>
> W3. **Does the introduction of the transformer-based bi-directional encoder and the mean-pooled representation compromise the model's causality?**
>
> No, it doesn't. **Each token cannot access the chunk it belongs to**, it can only access the previous chunks. We will make further clarification in the revised version.
>
>
> W5. **Can HSA operate independently of the Mamba blocks? If a model were designed using only HSA, would it still be faster than NSA?**
>
> We believe so. It can work together with sliding window attention. We believe HSA would still be faster than NSA because we do not require token compression attention, thus the overall FLOPs are lower.
>
> Q1. **Why does HSA adopt the stick-breaking attention mechanism for constructing chunk weights, rather than other forms of attention with positional encoding? I would like to see a theoretical justification or supporting ablation studies.**
>
> If we use RoPE, a straightforward issue would be the out-of-domain problem during extrapolation (when the input length exceeds the pretraining length), as discussed in [4] and [7].
>
> Here is an ablation study to replace the stick-breaking mechanism with softmax+RoPE:
>
> ppl with context length=64K  (the lower the better):
> | Models | pg19 | arxiv | code |
> |-|-|-|-|
> RAMba stick-breaking | 17.01| 3.65 | 3.07 |
> RAMba rope + softmax | 17.80 | 4.19 | 3.32 |
>
> RULER: (context len=64K) (the higher the better)
> | Models | S-N | MQ-N | VT | FWE|
> |-|-|-|-|-|
> RAMba stick-breaking | 85.07 | 55.22 | 55.22 | 53.73 |
> RAMba rope + softmax | 43.29 | 26.87 | 29.85 | 37.31 |
>
>
> Q2. **Does the equation for the HSA layer in this line omit the Mamba layers?**
>
> No, sorry for the confusion.
> It doesn't omit the Mamba layers.
> The layer indices here include both Mamba and HSA layers, and we will clarify this point further later.
>
> Q3. **Why does NSA's implementation currently not support generation?**
>
> As there is no official implementation for NSA thus we use an implementation from the open-source community (line 230).
>
> NSA's batch forward mode (used for training) calculates next-token perplexity for full sequences without needing KV cache management for new tokens, as no new tokens are generated. In contrast, generation mode requires handling a variable number of newly generated tokens, complex kv cache management, and positional encoding, especially with different attention mechanisms. This complexity is why vLLM performs well for inference but not training, while Megatron is optimized for the reverse. The KV management for generation was incomplete at the time of the paper's submission.
>
> Even so, this does not undermine our conclusions. The majority of experiments can be evaluated via batch forward passes, and the performance on PPL and RULER is sufficient to support our claims.
>
> [1] Jamba: Hybrid Transformer-Mamba Language Models
>
> [2] SAMBA: SIMPLE HYBRID STATE SPACE MODELS FOR EFFICIENT UNLIMITED CONTEXT LANGUAGE MODELING
>
> [3] Understanding and Improving Length Generalization in Recurrent Models
>
> [4] YaRN: Efficient Context Window Extension of Large Language Models
>
> [5] Repeat After Me: Transformers are Better than State Space Models at Copying
> Transformers are Better than State Space Models at Copying
>
> [6] MiniMax-01: Scaling Foundation Models with Lightning Attention
>
> [7] Length Generalization of Causal Transformers without Position Encoding
>
> We hope our replies address your concerns, and we would appreciate it very much if you could give us more support.

---

> > ### Comment · Reviewer_XZCu · 2025-08-03
> >
> > Thanks for the response. In particular, the example illustrating the advantage of HSA over NSA in addressing the inaccurate chunk selection problem is vivid, and I agree with reviewer eEPG’s suggestion to present this explanation as a figure or a more formal observation. As for HSA itself, it is another excellent work on sparse attention. At the same time, the authors should clearly explain certain potentially ambiguous phrases in the paper, such as “unable to randomly access historical context” and “inaccurate chunk selection.” In addition, given the number of components in the model, I found the Methods section to be somewhat compressed, with certain details insufficiently explained (as also pointed out by reviewer fVSb). The authors may consider expanding the explanations in the main text while moving more technical content—such as the forward and backward algorithms for the kernel—to the appendix. I consider raising my score.

---

> ### Author Response · Authors · 2025-08-04
>
> Thank you very much for your valuable suggestions and for recognizing the contributions of HSA.
>
> We greatly appreciate your feedback, particularly the suggestion to move the more technical content into the appendix, which we find highly helpful.  Given the positive feedback on the example, we plan to present it as a figure within the methodology section to better illustrate the distinction between NSA and HSA, replacing the original kernel design section. The kernel design section will then be relocated to the appendix. This adjustment not only enhances the overall flow but also provides clearer reasoning behind issues such as "inaccurate chunk selection."
>
> The space freed up by this reorganization will be utilized to address the ambiguous phrases you highlighted and to provide a more comprehensive explanation of the design rationales behind each component, with particular elaboration on the encoder.

---

### Official Review · Reviewer_eEPG · 2025-06-30

**Clarity:** 2
**Significance:** 3
**Originality:** 3
**Rating:** 4
**Confidence:** 4

**Summary:**

This paper introduces RAMba, which combines Mamba (a Linear RNN) with a novel attention method, called Hierarchical Sparse Attention (HSA), whose goal is to optimize the tradeoffs between efficiency, length generalization, and random access in long-context modeling.
The key innovation is HSA, a two-stage hierarchical attention mechanism that learns chunk-aware token-to-chunk relevance.
The authors implement hardware-aligned kernels for efficiency and introduce a memory reset mechanism during training to improve length generalization.
Ramba achieves perfect accuracy on context passkey retrieval up to 64M context length despite pre-training on only 4K contexts.
On language modeling tasks, it performs slightly outperforms with Transformers and Mamba on IID data (4K length), and performs on par with Mamba on OOD samples (16K and 64K context length).

**Questions:**

- Can you provide concrete evidence (citations, experiments) supporting the claim that existing chunk-based methods "suffer from inaccurate chunk selection"? I really think this is crucial because without this evidence, the fundamental motivation for HSA remains somewhat obscure.

- Do the PPUs used in efficiency evaluation support tensor cores? If not, I don't see how you can validly compare against FlashAttention-2, which is optimized specifically for tensor core utilization.

- Are the perplexity results in Table 1 on validation or test sets?

- How and where is the sorting operation applied? If it's inside the Triton kernel, how is it implemented?

- Can you provide a detailed breakdown of the FLOPs required for HSA and its arithmetic intensity? Specifically, I'd like to understand the computational cost per token including chunk selection, token-level attention, and chunk-level fusion, and whether your method is memory-bound or compute-bound.

**Ethical Concerns:**

["NO or VERY MINOR ethics concerns only"]

**Final Justification:**

Revised score after rebuttal: clarifications regarding the functionality of the method and its evaluation.

**Limitations:**

I think the authors provide a limitations section in the Appendix acknowledging the focus on Mamba and computational resource constraints. However, I believe some limitations could be addressed in order to improve the paper:

- Discussion of how findings might change at larger model/context scales

- I don't see any acknowledgment of the experimental inconsistencies or hardware evaluation issues that concern me (see weakness above)

- I believe there's limited theoretical analysis of when/why the approach might fail.

**Paper Formatting Concerns:**

Table captions/titles appear after instead of before the table.

**Quality:**

3

**Strengths And Weaknesses:**

**Strengths**

- I believe the hierarchical sparse attention mechanism represents a meaningful advance over existing chunk-based methods. The distinction between "chunk-aware" and "chunk-unaware" learning seems intuitive and well-motivated to me, with the two-stage attention design enabling end-to-end optimization of chunk selection.

- The 64M context passkey retrieval with perfect accuracy is quite impressive. That's 16000x beyond pre-training length, which I think demonstrates exceptional length extrapolation. The consistent improvements across language modeling benchmarks within their evaluated scope also support the method's effectiveness.

- I appreciate that the authors provided hardware-aligned kernel design with detailed algorithmic descriptions (Algorithms 1-3), which is very important for reproducibility and practical deployment.

- The efficiency-generalization-access trilemma is well-articulated and represents a fundamental challenge in long-context modeling.

**Weakness**

While I largely enjoying reading this paper, I believe it still has room for improvement. Here are some of them:

- The paper states that existing methods "often suffer from inaccurate chunk selection" (Line 37) without citing any supporting evidence. I think this unverified claim serves as the primary motivation for their approach, which I consider a significant methodological flaw.

- I believe the multiple interacting components (Mamba + HSA + chunk encoder + memory reset) make it difficult to isolate individual contributions and understand failure modes. A more objective presentation and evaluation of HSA sounds more compelling.

- The paper reports NSA results for language modeling (Table 1), passkey retrieval (Figure 3), and other tasks using Mamba with NSA, but later states "NSA's implementation currently does not support generation." Since language modeling IS generation (next-token prediction), I think this reveals either implementation inconsistencies or conceptual confusion (possibly on my side).

- I find it hard to accept the claim that Mamba "excels" (Line 292) at RULER tasks based on 20-40% accuracy rates. I'd say, instead, that all models perform poorly after a context length of 64K.

- I believe that a critical issue lies in the efficiency comparison using "Physics Processing Units (PPUs)", which, to the best of my knowledge, lack tensor cores. In turn, this makes me think the comparisons with FlashAttention-2 (specifically optimized for tensor core utilization) are misleading, making the efficiency claims invalid or overstretched.

- Finally, the paper lacks detailed computational complexity analysis. Based on my calculations (disregarding sorting), for the settings from section 4.1 (K=8 chunks, S=64 chunk size), HSA requires approximately $O(n \times \lceil n/S \rceil + n \times K \times S^2)$ FLOPs per sequence. For a 16K sequence with $d=1024$, we get ~540M FLOPs per token, and an arithmetic intensity of ~33 FLOPs/byte. This indicates that HSA is likely compute-bound rather than memory-bound on modern hardware (e.g., GPUs), which could better elucidate how memory offloading can provide empirical benefits.

---

> ### Author Rebuttal · Authors · 2025-07-27
>
> We sincerely appreciate your insightful feedback and are delighted to hear that you largely enjoyed reading our paper.
>
> W1, Q1: **Can you provide concrete evidence supporting the claim that existing chunk-based methods "suffer from inaccurate chunk selection"?**
>
> Apologies for the confusion.
>
> In line 38, "a closer examination reveals a critical weakness" refers to the evidence observed in our experiments. The chunk-based methods mentioned in our introduction mainly refer to NSA, which we consistently compare against in all experiments. Notably, in Table 3, Mamba(w/ NSA) performs significantly worse than RAMba(w/o enc) on the more challenging MQ-NIAH task, for in-domain sequence lengths. The gap becomes even more pronounced in length extrapolation. Please note these two models only differ in the attention module (NSA vs HSA).
>
> The results of RAMba(w/ NSA, w/ m.r.) further support the claim. Without RNN's memory, where retrieval relies solely on NSA, there is a significant performance drop across all tasks. In contrast, RAMba(w/ m.r.) exhibits no such issue.
>
> Here, **please allow us to further explain through theory** and specific examples why NSA may suffer from inaccurate chunk selection, as well as how HSA helps mitigate this problem.
>
> The core principle of NSA's learnable chunk selection lies in approximating the token-to-chunk relevance using unnormalized attention scores. In self-attention, the relevance of token j to token i is defined as:
>
> p[i,j] = $\frac{e^{logits_{i,j}}}{Z_i}, \quad Z_i = \sum_{j < i}{e^{logits_{i,j}}}$,
>
> where logits[i,j] = $q_i^T k_j$ is the dot product between the query $q_i$ of token i and the key $k_j$ of token j.
>
> The relevance of chunk c to token i is ideally the sum of the relevance of all tokens within that chunk:
>
> r[i,c] = $\sum_{j \in \mathcal{C}}{p_{i,j}}= \frac{1}{Z_i} \sum_{j \in \mathcal{C}} e^{logits_{i,j}}$,
>
> where $\mathcal{C}$ denotes tokens in chunk c.
> However, calculating r[i,c] requires computing $Z_i$ (the full softmax normalization across all tokens), which would necessitate full $ Q^TK$ computation and thus undermine the computational efficiency.
>
> To ensure efficiency, NSA approximates chunk relevance using the mean-pooled key representation $K_c$ of chunk c:
>
> r'[i,c] = $q_i^T K_c= q_i^T \frac{1}{S} \sum_{j \in \mathcal{C}}{k_j} = \frac{1}{S} \sum_{j \in \mathcal{C}} q_i^T k_j = \frac{1}{S} \sum_{j \in \mathcal{C}}{logits_{i,j}}$,
>
> where $K_c=\frac{1}{S} \sum_{j \in \mathcal{C}}{k_j}$. While this approximation bypasses the need for normalized softmax scores across all tokens, it introduces a discrepancy between r'[i,c] (the approximation) and r[i,c] (the ideal one).
>
> Let's consider the following example, assuming every 2 tokens form a chunk, separated by |:
> ```
> chunk ids       |       0      |      1    |      2    |     3     | ...
> token positions |   0   |  1   |  2  |  3  |  4  |  5  |  6  |  7  | ... , i
> logits[i,j]     | -20.0 | 20.0 | 1.0 | 0.0 | 2.0 | 1.0 | 1.0 | 1.0 |
> p[i,j]          |   0.0 |  1.0 | 0.0 | 0.0 | 0.0 | 0.0 | 0.0 | 0.0 |
> r[i,c]          |      1.0     |    0.0    |    0.0    |    0.0    |
> r'[i,c]         |      0.0     |    0.5    |    1.5    |    1.0    |
> ```
>
> If only the top-2 chunks can be selected, NSA would choose chunks 2 and 3 according to r' [i,c], thus missing the chunk 0 with the highest sum of attention weights.
>
> Assuming NSA selects the top-2 chunks for sparse attention, the following process would be:
> ```
> chunk ids         |      2      |      3      |
> token positions   |   4  |   5  |   6  |   7  |
> logits[i,j]       |  2.0 |  1.0 |  1.0 |  1.0 |
>                   |<-----     softmax   ----->|
> p'[i,j]           | 0.49 | 0.17 | 0.17 | 0.17 |
> ```
>
> Throughout this process, r'[i,c] only participates in chunk selection but not in the forward computation, nor does it receive gradients.
>
> **The NSA's inaccurate chunk selection issue stems from using unnormalized attention logits to estimate chunk importance, with r'[i,c] not learnable, making the inaccuracy of chunk selection unavoidable.**
>
> For the same example, HSA works as follows:
> ```
> chunk ids          |       0      |      1     |      2    |     3     | ...
> token positions    |   0   |   1  |  2   |  3  |  4  |  5  |  6  |  7  | ...
> logits[i,j]        | -20.0 | 20.0 |  1.0 | 0.0 | 1.0 | 1.0 | 1.0 | 1.0 | ...
> p[i,j]             |  0.0  |  1.0 |  0.0 | 0.0 | 0.0 | 0.0 | 0.0 | 0.0 | ...
> s[i,c]             |    s [i,0]   |  s [i,1]   |  s [i,2]  |  s [i,3]  | ...
> ```
> Where s [i,c] represents learnable token-to-chunk relevance scores as described in Line 155 of the paper. For example, HSA may also select chunks 2 and 3 when parameters are randomly initialized, we have:
> ```
> chunk ids            |           2           |          3          |
> token positions      |     4     |     5     |     6    |     7    |
> attn_logits [i,j]    |   2.0     |    1.0    |    1.0   |    1.0   |
> intra-chunk weighting|<--     softmax      ->|<--    softmax     ->|
> token-level weights  |   0.73    |    0.27   |    0.5   |    0.5   |
> s[i,c]               |        s[i,2]         |        s[i,3]       |
> inter-chunk weighting|<---            stick-breaking           --->|
> chunk-level weights  |        w[i,2]         |        w[i,3]       |   // s.t. w [i,2} + w [i,3] <= 1
> final weights        |0.73×w[i,2]|0.27×w[i,2]|0.5×w[i,3]|0.5×w[i,3]|
> ```
>
> In this process, w[i,c], derived from s[i,c], participates in the final attention weight computation and the whole forward pass. This allows it to receive gradients and allocate higher weights to chunks with more important tokens. Even if random initialization initially misses the most relevant chunk 0, continuous training enables the model to gradually learn to select the most relevant chunks.
>
> This point can be further validated through a pure NSA Transformer.
> RULER results:
> | Models | 4K | S-N | MQ-N | VT | FWE |
> |-|-|-|-|-|-|
> |Transformer full-attn| | 95.08 | 88.59 | 97.12 | 60.0 |
> | Transformer w/ NSA | | 46.85 | 5.66 | 19.48 | 51.95 |
>
>
> W2. **The multiple interacting components make it difficult to isolate individual contributions**
>
> We fully agree with your perspective and have conducted ablation studies on each component to observe their individual impacts.
>
> Table 3 includes all ablation experiments, such as removing the encoder (w/o enc), memory reset (w/o m.r.), and replacing HSA with NSA or purely MAMBA, reflecting the influence of different modules on the overall system.
>
> The most straightforward comparison is between RAMba(w/o enc), Mamba, and Mamba(w/ NSA). The only difference lies in HSA, which shows significant improvement for inputs of 64K and below. The other variants, such as memory reset and encoder, primarily serve to further enhance HSA's extrapolation capability.
>
> W3. **NSA's implementation currently does not support generation.**
>
> As there is no official implementation for NSA thus we use an implementation from the open-source community (line 230).
>
> NSA's batch forward mode (used for training) calculates next-token perplexity for full sequences without needing KV cache management for new tokens, as no new tokens are generated. In contrast, generation mode requires handling a variable number of newly generated tokens, complex kv cache management, and positional encoding, especially with different attention mechanisms. This complexity is why vLLM performs well for inference but not training, while Megatron is optimized for the reverse. The KV management for generation was incomplete at the time of the paper's submission.
>
> Even so, this does not undermine our conclusions. The majority of experiments can be evaluated via batch forward passes, and the performance on PPL and RULER is sufficient to support our claims.
>
> W4. **I find it hard to accept the claim that Mamba "excels".**
>
> Thanks for pointing that out. The original phrasing was imprecise. We will revise it to: "Mamba performs better in long sequential statistical tasks."
>
> W6/Q5. **How memory offloading can provide empirical benefits. Regarding computational complexity analysis.**
>
> Sorry for the confusion, the "memory offloading" refers to offloading the KV cache from **GPU to the CPU**, not HBM to SRAM; Memory offloading does not provide computational benefits; in fact, it slows down generation but reduces the GPU's memory footprint, as reflected in Figure 4(3) and Table 7.
> During long-sequence inference, the KV cache could overwhelm GPU memory. Offloading the KV cache to CPU and just reloading the KV cache of selected chunks can largely save the memory footprint.
> Therefore, memory offloading has no relationship with whether HSA is memory-bound or compute-bound.
>
> The complexity of HSA is as follows:
>
> L: sequence length, d: hidden size, K: number of selected chunks, S: number of tokens in a chunk
> Retrieval: $O(L \times \frac{L}{S} \times d)$
>
> Attention: $O(L \times K \times S \times d)$
> Compared to your version, there is no $S^2$, each token attends to $K \times S$ tokens.
>
> Chunk-level fusion: $O(L \times K \times d)$
>
> Q2: **Do the PPUs support tensor cores?**
>
> Yes, they do.
> Here, the "PPU" is just another name for GPUs not produced by NVIDIA. It also supports Torch and Triton.
> Recently, we have access to the H800, and the throughput of 8*H800 is twice the results shown in Table 6, which aligns with Figure 4. Without tensor cores, the PPU could not achieve such throughput.
>
> Q3. **Are the perplexity results on validation or test sets?**
>
> On validation sets.
>
> Q4. **How and where is the sorting operation applied?**
>
> When computing token-to-chunk weighting in line 155. It's outside the HSA triton kernel; weights and indices are already computed before conducting HSA.
>
> Your suggestion on discussing the chunk selection issue is very helpful, and we will include it in the Appendix. We hope our replies address your concerns, and we would appreciate it very much if you could consider raising the score.

---

> > ### Comment · Reviewer_eEPG · 2025-08-03
> >
> > Thank you for your comprehensive response.
> >
> > Thank you also for the excellent example; it is extremely helpful! I **strongly** encourage you to turn it into a figure and include it in the appendix.
> >
> > Some follow-up comments below:
> >
> > **"[...] This allows it to receive gradients and allocate higher weights to chunks with more important tokens."**
> >
> > I'm not sure this is necessarily beneficial, as it introduces an additional gradient path that updates the query and key weight matrices. Have you observed any instabilities during training as a result of this?
> >
> > **""This point can be further validated through a pure NSA Transformer."**
> >
> > This actually relates to one of my earlier concerns. I believe HSA is a strong contribution on its own. Combining it with Mamba all at once introduces a lot of complexity and can be confusing, especially with further hacks such as the "memory reset" approach.
> >
> > That said, I still find the overall work very compelling; it just makes the paper harder to follow.
> >
> > **"NSA's implementation currently does not support generation."**
> >
> > Agreed. A more careful choice of wording here could help clarify this point.
> >
> > **""The complexity of HSA"**
> >
> > Could you expand on this in terms of the number of reads, writes, and FLOPs?
> >
> >
> > PS: I've updated my score to reflect the improvements and clarifications provided so far.

---

> > > ### Author Response · Authors · 2025-08-03
> > >
> > > Thank you very much for your prompt reply.
> > >
> > > **Have you observed any instabilities during training as a result of this?**
> > >
> > > Currently, we are scaling a RAMba-based LLM and have trained 4T tokens. From empirical results, the training process is stable overall.
> > >
> > > **Could you expand on this in terms of the number of reads, writes, and FLOPs?**
> > >
> > > Of course, we’d be happy to.
> > >
> > > We implemented the following code to estimate FLOPs and reads/writes for each token:
> > >
> > > ```python
> > > def compute_hsa_per_step(
> > >     batch_size,
> > >     L,
> > >     chunk_size,
> > >     top_k,
> > >     hidden_size,
> > >     retrieval_dim,
> > >     num_key_value_heads
> > > ):
> > >     # (B, 1, d) @ (d, d) -> 2 * B * L * d^2, 2 for dot-products and summations
> > >     q_proj_flops = 2 * batch_size * 1 * hidden_size**2
> > >     # chunk selection (B, 1, d) @ (d, L // chunk_size) -> 2 * B * L * d
> > >     retrieval_flops = 2 * batch_size * num_key_value_heads * 1 * retrieval_dim * (L // chunk_size)
> > >
> > >     # attn dot product + weighted sum
> > >     attn_flops = (
> > >         # inter-chunk attention (B, 1, d) @ (d, K * chunk_size)
> > >         2 * batch_size * 1 * top_k * chunk_size * hidden_size +
> > >         # intra-chunk attention (B, 1, k) @ (k, d)
> > >         2 * batch_size * 1 * top_k * hidden_size +
> > >         # output projection (B, 1, d) @ (d, d)
> > >         2 * batch_size * 1 * hidden_size**2
> > >     )
> > >
> > >     total_flops = q_proj_flops + retrieval_flops + attn_flops
> > >     # backward is around 2 times the forward
> > >
> > >     # chunk size (K * S * d)
> > >     read_floats = (
> > >         # 16 q heads share a query head
> > >         batch_size * 1 * top_k * chunk_size * hidden_size // 16 +
> > >         batch_size * 1 * num_key_value_heads * retrieval_dim * (L // chunk_size)
> > >     )
> > >     write_floats = (
> > >         # output final O
> > >         batch_size * 1 * hidden_size +
> > >         # output chunk weights
> > >         batch_size * 1 * num_key_value_heads * top_k
> > >     )
> > >
> > >     return total_flops, read_floats, write_floats
> > > ```
> > >
> > > For different context length (256, 2048, 16K), with d=1024, retrieval_dim=1024, one retrieval head, we have:
> > > ```python
> > > print(compute_hsa_per_step(1, 256, 64, 8, 1024, 1024, 1))
> > > print(compute_hsa_per_step(1, 2048, 64, 8, 1024, 1024, 1))
> > > print(compute_hsa_per_step(1, 16384, 64, 8, 1024, 1024, 1))
> > > ```
> > >
> > >
> > > The results are as follows:
> > >
> > > (5267456, 36864, 1032)
> > >
> > > (5324800, 65536, 1032)
> > >
> > > (5783552, 294912, 1032)
> > >
> > > If we set retrieval dim to 256:
> > > ```python
> > > print(compute_hsa_per_step(1, 256, 64, 8, 1024, 256, 1))
> > > print(compute_hsa_per_step(1, 2048, 64, 8, 1024, 256, 1))
> > > print(compute_hsa_per_step(1, 16384, 64, 8, 1024, 256, 1))
> > > ```
> > > we have results:
> > >
> > > (5261312, 33792, 1032)
> > >
> > > (5275648, 40960, 1032)
> > >
> > > (5390336, 98304, 1032)

---

### Official Review · Reviewer_hWyn · 2025-07-06

**Clarity:** 3
**Significance:** 4
**Originality:** 3
**Rating:** 5
**Confidence:** 4

**Summary:**

This paper introduces a novel Hierarchical Sparse Attention (HSA) mechanism to give RNN-based models like Mamba random-access capability while maintaining linear complexity. By integrating HSA with a hardware-aligned kernel and memory offloading, the resulting RAMba model shows exceptional length generalization and efficiency. Notably, it achieves perfect accuracy on a 64M-token passkey retrieval task despite being trained on only a 4K context.

**Questions:**

1.	Regarding the model’s architectural complexity and its new hyperparameters (S, K, G): could the authors comment on their sensitivity? How were the reported values chosen, and what is the expected impact of varying them?
2.	Regarding the memory reset mechanism: could the authors provide an ablation study against simpler strategies (e.g., resetting to zero or to the last state) to better justify the choice of a randomized approach?
3.	To validate the true advantage of the HSA hierarchical design, we suggest the authors provide results for an NSA baseline augmented with the same chunk encoder. This experiment is crucial for a fair comparison.

**Ethical Concerns:**

["NO or VERY MINOR ethics concerns only"]

**Limitations:**

yes.

**Paper Formatting Concerns:**

No.

**Quality:**

4

**Strengths And Weaknesses:**

Strengths:
1.	The paper’s core contribution is a novel, well-motivated Hierarchical Sparse Attention (HSA) mechanism that directly addresses the critical trade-off between efficiency and random-access capability in long-context models.
2.	The claims are backed by extensive and compelling experiments across a wide range of tasks. The standout performance on a 64M-token passkey retrieval task strongly validates the method’s length extrapolation capabilities.
3.	The work extends beyond theory to a practical, system-level solution, including a hardware-aligned kernel design and an efficient inference strategy, which significantly enhances its real-world applicability.
Weaknesses:
1.	While powerful, the RAMba architecture introduces significant complexity and multiple new hyperparameters (e.g., chunk size S, selected chunks K, HSA layer frequency G). The paper lacks ablation studies on these crucial design choices, making it difficult to assess their individual impact or the model’s sensitivity.
2.	The choice of a randomized segment for the memory reset mechanism is not sufficiently justified. Although shown to be effective, the lack of comparison against simpler strategies (like resetting to zero) makes it unclear whether the randomization itself is a key ingredient or an unnecessary complexity.
3.	The comparison against the NSA baseline is confounded. As shown in Fig. 3, RAMba has a significant advantage over RAMba-w/o-enc, which proves the chunk encoder is critical. Given this, the comparative experiments should have equipped the NSA baseline with the same component to fairly test the merits of the hierarchical attention design itself. Without this control, the source of the performance gain remains ambiguous.

---

> ### Author Rebuttal · Authors · 2025-07-28
>
> We sincerely appreciate your insightful feedback and recognition of our work.
>
> W1,Q1. **Regarding the model’s architectural complexity and its new hyperparameters (S, K, G): could the authors comment on their sensitivity? How were the reported values chosen, and what is the expected impact of varying them?**
>
> Based on previous research on hybrid models ([1], Fig.4), the optimal ratio between attention and Mamba is generally around 8.25\%~20\%. Therefore, we inserted 3 HSA layers into the 48-layer Mamba model. Generally, S is set to 64 to better align with hardware.
>
> The value of K primarily impacts computational cost—larger K values typically result in better performance but also higher computational load and lower sparsity. The optimal value of K is worth conducting an empirical study on. However, since this work requires comparison with other baselines to verify its effectiveness, we have not delved into this aspect here. This will be left for future work.
>
> W2,Q2. **Regarding the memory reset mechanism: could the authors provide an ablation study against simpler strategies (e.g., resetting to zero or to the last state) to better justify the choice of a randomized approach?**
>
> Thank you for your insightful question regarding the memory reset mechanism. To address your concern, we would like to clarify that our approach involves initializing the memory state using the last hidden state from a randomly selected segment.
>
> In fact, a recent study [2] has conducted an empirical analysis specifically on this topic.
> They refer to using the last hidden state from a random segment as the initial state as "state passing," and using the last hidden state from the preceding sentence as the initial state as "truncated backpropagation through time (BPTT or TBTT). They compared different strategies, including setting to zero, TBTT, and state passing.
> Their results ([2], Fig. 5) demonstrate that initializing the initial state with the last state from a random segment (as employed in our paper) consistently outperforms other alternatives when extrapolating, such as resetting to zero or initializing with a randomized vector. This evidence supports the effectiveness of our chosen approach.
>
> W3, Q3. **We suggest the authors provide results for an NSA baseline augmented with the same chunk encoder**
>
> We sincerely appreciate your suggestion. We did not conduct the comparison for two reasons. First, we believe that RAMba (w/o enc) is a fair comparison with Mamba w/ NSA. Secondly, NSA is inherently incompatible with the chunk encoder, as it is unique to the hierarchical attention framework. Please allow us to elaborate further.
>
> One purpose of introducing RAMba (w\o enc) is to enable a fair comparison with NSA. Aside from the differences in the attention module, RAMba (w\o enc) and Mamba (w\ NSA) are identical in every other aspect. However, RAMba (w\o enc) significantly outperforms Mamba (w\ NSA) on RULER (Table. 3), and this comparison validates that the performance improvement mainly stems from HSA.
>
> In addition, NSA itself is not compatible with the encoder. Let's take an example to see the difference between how NSA and HSA work.
>
> Assuming every 2 tokens form a chunk, separated by |:
> ```
> chunk ids             |       0      |      1    |      2    |     3     | ...
> token positions       |   0   |  1   |  2  |  3  |  4  |  5  |  6  |  7  | ... , i <- the currrent token
> attn logits[i, *]     | -20.0 | 20.0 | 1.0 | 0.0 | 2.0 | 1.0 | 1.0 | 1.0 |
> attn weights (softmax)|   0.0 |  1.0 | 0.0 | 0.0 | 0.0 | 0.0 | 0.0 | 0.0 |  // O(L$^2$), L: sequence length
> chunk attn logits sum |      0.0     |    0.5    |    1.5    |    1.0    |  // O(L*(L/S), S: chunk size
> ```
> NSA selects chunks according to "chunk attn logits sum" for efficiency.
> If only the top-2 chunks can be selected, NSA would choose chunks 2 and 3, thus missing the chunk 0 with the highest sum of attention weights, **leading to inaccurate chunk selection**.
>
> Assuming NSA selects the top-2 chunks for sparse attention, the following process would be:
> ```
> chunk ids         |      2      |      3      |
> token positions   |   4  |   5  |   6  |   7  |  // sentence-level RoPE applied here (0 ~ L)
> attn logits       |  2.0 |  1.0 |  1.0 |  1.0 |
>                   |<-   enc?  ->|<-   enc?  ->|
>                   // Chunk-wise encoding here will disrupt global positional information.
>                   |<-----     softmax   ----->|
> attn weights      | 0.49 | 0.17 | 0.17 | 0.17 |
> ```
> If chunk encoding is applied at the chunk level, it will destroy the global positional information.
>
> For the same example, HSA works as follows:
> ```
> chunk ids          |       0      |      1     |      2    |     3     | ...
> token positions    |   0   |   1  |  2   |  3  |  4  |  5  |  6  |  7  | ...
> attn logits        | -20.0 | 20.0 |  1.0 | 0.0 | 1.0 | 1.0 | 1.0 | 1.0 | ...
> s[i,c]             |    s [i,0]   |  s [i,1]   |  s [i,2]  |  s [i,3]  | ...
> ```
> Where s [i,c] represents learnable token-to-chunk relevance scores as described in Line 155 of the paper. For example, HSA may also select chunks 2 and 3 when parameters are randomly initialized, we have:
> ```
> chunk ids            |           2           |          3          |
>                      |<---      enc      --->|<---      enc    --->| // chunk-wised encoding
> attn logits          |   2.0     |    1.0    |    1.0   |    1.0   | // intra-chunk RoPE only (0~S for each chunk)
> intra-chunk weighting|<--     softmax      ->|<--    softmax     ->|
> token-level weights  |   0.73    |    0.27   |    0.5   |    0.5   |
> s[i,c]               |        s[i,2]         |        s[i,3]       |
> inter-chunk weighting|<---            stick-breaking           --->| // with positional inductive bias
> chunk-level weights  |        w[i,2]         |        w[i,3]       |   // s.t. w [i,2} + w [i,3] <= 1
> final weights        |0.73×w[i,2]|0.27×w[i,2]|0.5×w[i,3]|0.5×w[i,3]|
> ```
> Due to the positional inductive bias inherent in the stick-breaking attention mechanism, it becomes feasible to employ a bidirectional encoder at the chunk level.
>
> Moreover, in this process, w[i,c], derived from s[i,c], participates in the final attention weight computation and the whole forward pass. This allows it to receive gradients and allocate higher weights to chunks with more important tokens. Even if random initialization initially misses the most relevant chunk 0, continuous training enables the model to gradually learn to select the most relevant chunks.
>
>
> [1] An Empirical Study of Mamba-based Language Models
>
> [2] Understanding and Improving Length Generalization in Recurrent Models

---

### Official Review · Reviewer_fVSb · 2025-07-13

**Clarity:** 2
**Significance:** 3
**Originality:** 2
**Rating:** 3
**Confidence:** 4

**Summary:**

This paper introduces Hierarchical Sparse Attention (HSA), a novel attention mechanism designed to provide RNNs with efficient, long-range random access capabilities, addressing the 'trilemma' of balancing efficiency, random-access flexibility, and length generalization.

HSA operates by dividing the input into chunks, selecting the most relevant chunks, and then applying a two-stage attention process: first within tokens of a chunk, and then across the selected chunks. The authors integrate HSA with a Mamba-2 backbone to create RAMba. To manage memory, RAMba employs techniques like hardware-aligned kernels and offloading the KV cache to CPU memory.

The paper's experiments demonstrate that RAMba achieves near state-of-the-art performance on long-context tasks, such as perfect accuracy on a 64-million-token passkey retrieval task , while maintaining high throughput and a near-constant memory footprint during inference.

**Questions:**

1. Can the authors also reports the results for pure NSA w/o SWA as used in MiniCPM4? I think it is an important baseline besides transformer.
2. Could you elaborate more on BPTT? How did you process the last hidden states after param optimized, If I understand correctly, the saved states does not align with the ones produced by the updated model weights, will this hurt the performance?
3. I did not figure out the difference between the alg in  Fig. 2 (b) and Algorithm 1 with NSA.
4. How about the performance of MK-NIAH on RULER?
5. I think some more results beyond RULER and long-bench could be present to verify the claims of random long-context access abilities. Wondering if it's possible to report the results of Phonebook lookup used in "Repeat After Me", which is training-free

MiniCPM4: https://arxiv.org/abs/2506.07900

Repeat After Me: https://arxiv.org/pdf/2402.01032

**Ethical Concerns:**

["NO or VERY MINOR ethics concerns only"]

**Limitations:**

yes

**Quality:**

3

**Strengths And Weaknesses:**

RAMba is a novel architecture that integrates HSA with a Mamba backbone. The goal is to equip Mamba with efficient, long-range random access capabilities, a direction of significant interest for sequence modeling. While the approach is promising and the results are strong, I have a few concerns regarding the architectural design and the justification for its components.

* On the Source of Performance Gains in HSA: A central claim of HSA is its "chunk-aware" learning mechanism, which is presented as its core innovation over prior work. However, the mechanics of how this is achieved beyond standard backpropagation through the hierarchical structure could be elaborated upon. This point is crucial because the ablation results in Table 1 for RAMba w/o s.b. show a significant performance degradation when stick-breaking attention is removed. This raises the question of whether the primary performance gains stem from the specific properties of the stick-breaking mechanism itself, rather than the proposed two-stage hierarchical attention. Further clarification or a more targeted ablation could help disentangle these effects and solidify the paper's core claim.
* Justification for the Transformer-based Encoder: The choice of a Transformer-based bi-directional encoder for chunk memory is another key design decision. What was the specific motivation for this choice over other alternatives, such as using another Mamba layer? Given that RAMba is a Mamba-centric architecture, integrating a Transformer component introduces additional complexity. An ablation study comparing different encoder designs would significantly strengthen the justification for this specific architectural choice and its contribution to the model's impressive length generalization capabilities.

Generally, I think the the overall design of RAMba is quite sophisticated, with several novel components (HSA, chunk encoder, memory reset) interleaved with the Mamba backbone.
While the results on very long contexts are impressive, a discussion on the model's practical efficiency (latency and throughput) on shorter sequences—which are common in many real-world scenarios—would be beneficial. It is unclear how the overhead from the added components impacts performance at smaller scales.

To make the contribution more convincing and the design principles clearer, the authors could add a more detailed architectural justification. For instance, explaining the rationale behind alternating one HSA layer with G Mamba layers in the upper decoder could help readers understand the intended synergy between the components. A more minimalist design, supported by careful ablations that justify each component's inclusion, would make the final architecture even more compelling.

---

> ### Author Rebuttal · Authors · 2025-07-27
>
> We sincerely appreciate your insightful feedback.
>
> Q3,W1. **I did not figure out the difference between HSA and NSA.  How "chunk-aware" learning is achieved beyond standard backpropagation through the hierarchical structure could be elaborated upon**
>
> This is the most important contribution of this work, so please allow us to explain it first.
>
> Why NSA suffer from inaccurate chunk selection, and why can HSA learn from back-propagation?
>
> The core principle of NSA's learnable chunk selection lies in approximating the token-to-chunk relevance **using unnormalized attention scores**. In self-attention, the relevance of token j to token i is defined as:
>
> p[i,j] = $\frac{e^{logits_{i,j}}}{Z_i}, \quad Z_i = \sum_{j < i}{e^{logits_{i,j}}}$,
>
> where logits[i,j] = $q_i^T k_j$ is the dot product between the query $q_i$ of token i and the key $k_j$ of token j.
>
> The relevance of chunk c to token i is ideally the sum of the relevance of all tokens within that chunk:
>
> r[i,c] = $\sum_{j \in \mathcal{C}}{p_{i,j}}= \frac{1}{Z_i} \sum_{j \in \mathcal{C}} e^{logits_{i,j}}$,
>
> where $\mathcal{C}$ denotes tokens in chunk c.
> However, calculating r[i,c] requires computing $Z_i$ (the full softmax normalization across all tokens), which would necessitate full $ Q^TK$ computation and thus undermine the computational efficiency.
>
> To ensure efficiency, NSA instead approximates chunk relevance using the mean-pooled representation of keys:
>
> r'[i,c] = $q_i^T K_c= q_i^T \frac{1}{S} \sum_{j \in \mathcal{C}}{k_j} = \frac{1}{S} \sum_{j \in \mathcal{C}} q_i^T k_j = \frac{1}{S} \sum_{j \in \mathcal{C}}{logits_{i,j}}$,
>
> where $K_c$ represents the mean-pooling of key representations within chunk c. While this approximation bypasses the need for normalized softmax scores across all tokens, it introduces a discrepancy between r'[i,c] (the approximation) and r[i,c] (the ideal one).
>
> Let's consider the following example, assuming every 2 tokens form a chunk, separated by |:
> ```
> chunk ids       |       0      |      1    |      2    |     3     | ...
> token positions |   0   |  1   |  2  |  3  |  4  |  5  |  6  |  7  | ... , i (current token)
> logits[i,j]     | -20.0 | 20.0 | 1.0 | 0.0 | 2.0 | 1.0 | 1.0 | 1.0 |
> p[i,j]          |   0.0 |  1.0 | 0.0 | 0.0 | 0.0 | 0.0 | 0.0 | 0.0 |
> r[i,c]          |      1.0     |    0.0    |    0.0    |    0.0    |
> r'[i,c]         |      0.0     |    0.5    |    1.5    |    1.0    |
> ```
>
> If only the top-2 chunks can be selected, NSA would choose chunks 2 and 3 according to r' [i,c], thus missing the chunk 0 with the highest sum of attention weights.
>
> Assuming NSA selects the top-2 chunks for sparse attention, the following process would be:
> ```
> chunk ids         |      2      |      3      |
> token positions   |   4  |   5  |   6  |   7  |
> logits[i,j]       |  2.0 |  1.0 |  1.0 |  1.0 |
>                   |<-----     softmax   ----->|
> p'[i,j]           | 0.49 | 0.17 | 0.17 | 0.17 |
> ```
>
> Throughout this process, r'[i,c] only participates in chunk selection but not in the forward computation, nor does it receive gradients.
>
> **The NSA's inaccurate chunk selection issue stems from using unnormalized attention logits to estimate chunk importance, with r'[i,c] not learnable, making the inaccuracy of chunk selection unavoidable.**
>
> For the same example, HSA works as follows:
> ```
> chunk ids          |       0      |      1     |      2    |     3     | ...
> token positions    |   0   |   1  |  2   |  3  |  4  |  5  |  6  |  7  | ...
> logits[i,j]        | -20.0 | 20.0 |  1.0 | 0.0 | 1.0 | 1.0 | 1.0 | 1.0 | ...
> p[i,j]             |  0.0  |  1.0 |  0.0 | 0.0 | 0.0 | 0.0 | 0.0 | 0.0 | ...
> s[i,c]             |    s [i,0]   |  s [i,1]   |  s [i,2]  |  s [i,3]  | ...
> ```
> Where s [i,c] represents learnable token-to-chunk relevance scores as described in Line 155 of the paper. For example, HSA may also select chunks 2 and 3 when parameters are randomly initialized, we have:
> ```
> chunk ids            |           2           |          3          |
> token positions      |     4     |     5     |     6    |     7    |
> attn_logits [i,j]    |   2.0     |    1.0    |    1.0   |    1.0   |
> intra-chunk weighting|<--     softmax      ->|<--    softmax     ->|
> token-level weights  |   0.73    |    0.27   |    0.5   |    0.5   |
> s[i,c]               |        s[i,2]         |        s[i,3]       |
> inter-chunk weighting|<---            stick-breaking           --->|
> chunk-level weights  |        w[i,2]         |        w[i,3]       |   // s.t. w [i,2} + w [i,3] <= 1
> final weights        |0.73×w[i,2]|0.27×w[i,2]|0.5×w[i,3]|0.5×w[i,3]|
> ```
>
> In this process, w[i,c], derived from s[i,c], participates in the final attention weight computation and the whole forward pass. **This allows it to receive gradients and allocate higher weights to chunks with more important tokens.** Even if random initialization initially misses the most relevant chunk 0, continuous training enables the model to gradually learn to select the most relevant chunks.
>
> Q1. **Can the authors also report the results for pure NSA w/o SWA as used in MiniCPM4?**
>
> Since MiniCPM4 had not been released at the time of submitting the paper, we did not include a comparison. However, this suggestion was very helpful. We tested it, and the results are as follows:
>
> Perplexity:
> |Models| 4K | pg19 | arxiv | code | 16K | pg19 | arxiv | code | 64K | pg19 | arxiv | code |
> |-|-|-|-|-|-|-|-|-|-|-|-|-|
> |Transformer full-attn| | 18.61 | 4.23 | 3.28 | | 539.15 | 199.42 | 62.17 | | >104 | >104 | 2865.51 |
> |RAMba w/o m.r. | |17.63 | 4.13 |3.21 | | 17.11| 3.81 |3.08 | | 17.11 |3.87 |3.21|
> |Transformer w/ NSA| | 18.43 | 4.18 | 3.24 | | 20.74 | 4.55 | 3.46 | | 23.61 | 5.66 | 4.02|
>
> RULER:
> | Models | 4K | S-N | MQ-N | VT | FWE | 64K | S-N | MQ-N | VT | FWE |
> |-|-|-|-|-|-|-|-|-|-|-|
> |Transformer full-attn| | 95.08 | 88.59 | 97.12 | 60.0 | | 0.00 | 0.00 | 0.00 | 0.00 |
> |Ramba w/o m.r | | 92.76 | 87.10 | 96.66 | 76.81 | | 55.22 | 29.85 | 68.66 | 8.96 |
> | Transformer w/ NSA | | 46.85 | 5.66 | 19.48 | 51.95 | | 0.00 | 0.00 | 0.00 | 7.46 |
>
> This result further validates our argument that the NSA has issues with chunk selection, even for in-domain length.
>
> W1. **Whether the primary performance gains stem from the stick-breaking mechanism itself, rather than HSA.**
>
> RAMba$_{w/o s.b}$ means using softmax without position encoding instead of the stick-breaking mechanism (please refer to Line 242). We believe that the performance degradation is due to the inability to distinguish the relative distance between chunks, rather than additional improvement brought by the stick-breaking mechanism.
>
> Regarding the extrapolation ability,  the experiments in the original stick-breaking attention paper[1] (Figure 6)  show that stick-breaking attention itself cannot resolve the NIAH extrapolation issue, which demonstrates that the extrapolation capability is not brought by the stick-breaking mechanism. Moreover, the stick-breaking mechanism is not applied to tokens, but only to chunks, as we mentioned above.
>
> In summary, stick-breaking is one component of the overall design, and it undoubtedly contributes to our results to some extent. However, the core contributions of this paper, the extrapolation capability and the high performance of sparse attention, are still primarily achieved by HSA.
>
> W2. **Justification for the Transformer-based Encoder.**
>
> Since the intra-chunk attention of HSA is conducted with each chunk separately,  an intuitive idea is to use a bi-directional encoder to enhance chunk-level representations. Please note that it doesn't break causality, as each token is not allowed to access the chunk it belongs to.
>
> Using bi-directional Mamba layers may be another choice. Here are the results for RULER:
>
> |Models| 4K | S-N | MQ-N | VT | FWE | 64K | S-N | MQ-N | VT | FWE |
> |-|-|-|-|-|-|-|-|-|-|-|
> |RAMba tfm enc | | 91.74 | 80.61 | 95.18 | 49.81 | | 85.07 | 55.22 | 55.22 | 53.73 |
> |RAMba mamba enc | | 92.25 | 78.45 | 94.28 | 45.35 | | 83.58 | 56.72 | 53.73 | 47.76 |
>
> The results are quite close, which validates that the choice of encoder has minimal impact.
>
> Q2. **Could you elaborate more on BPTT?**
>
> Existing parallelizable RNNs often suffer from Oversized States[3]. A common solution is to use BPTT (Backpropagation Through Time) for pretraining.
>
> Simply put, the last state from the previous step is used, and after gradient truncation, it becomes the initial state for the current step.
>
> This aligns with the recent work[4] by the author of Mamba, where they term this mechanism as 'state passing.'
>
> As demonstrated in the experiments in [4] (Fig. 5), state passing has a negligible impact on results but significantly enhances extrapolation capability.
>
> Q5. **Phonebook lookup**
>
> Here are the results:
>
> | Models | 4k | #entries=10 | #entries=50 | 16K | #entries=10 | #entries=20 | 64K | #entries=10 | #entries=20 |
> |-|-|-|-|-|-|-|-|-|-|
> | Transformers full-attn| | 87.76 | 77.46 | | 0.0 | 0.0 | | 0.0 | 0.0 |
> | Mamba | | 0.0 | 0.0 | | 0.0 | 0.0 | | 0.0 | 0.0 |
> | Mamba w/ SWA | | 0.01 | 0.0 | | 0.0 | 0.0 | | 0.0 | 0.0 |
> | Mamba w/ NSA | | 14.65 | 1.11 | | 5.94 | 4.08 | | 0.0 | 0.0 |
> | RAMba | | 87.01 | 71.52 | | 75.80 | 63.56 | | 52.24 | 35.82 |
>
> Q4. **MK-NIAH on RULER**
>
> The MQ-NIAH test in Table 3 covers the multi-key NIAH test. It contains 6 key-value pairs in the context. Moreover, the phonebook lookup task is a harder version of MK-NIAH.
>
> [1] Scaling Stick-Breaking Attention: An Efficient Implementation and In-depth Study
>
> [2] Retrieval-Pretrained Transformer: Long-range Language Modeling with Self-retrieval
>
> [3] Stuffed Mamba: Oversized States Lead to the Inability to Forget
>
> [4] Understanding and Improving Length Generalization in Recurrent Models
>
> Your suggestion to compare Transformers with the NSA is extremely helpful. We have provided the relevant data to address your concerns and hope our responses adequately resolve the issues. We would greatly appreciate it if you could reconsider and raise the score.

---

> ### Comment · Reviewer_fVSb · 2025-08-02
>
> Thank you for your very thoughtful responses, which have addressed most of my concerns.
> I also greatly appreciate the authors' hard work in recent days.
> However, I still have some reservations regarding the complexity of the design and how it scales.
> As such, I am maintaining my score of 3. I am, however, very much looking forward to the improved version of the HSA.

---

> ### Author Response · Authors · 2025-08-03
>
> Thank you very much for your prompt reply.
>
> Regarding scalability:
> 1. **Tables 6-7 & Figure 4** comprehensively demonstrate our model's scalability in terms of context length (**16K~64K**) and parameter size (**370M~3B**).
> 2. **Table 5** further reports results after continued pretraining on a **3B** model.
>
> If your concerns are about latency and throughput for short sequences, here are the results:
>
> **Per-step inference cost:**
> | Models | 64 | 256 | 1024 |
> |--------|----|-----|-----|
> | Mamba  | 3.06 | 2.98 | 3.04 |
> | RAMBA  | 3.2  | 3.26 | 3.21 |
>
> Even for short texts, the additional overhead introduced by HSA is less than 10% compared to vanilla Mamba.
>
> **Training throughputs at 2K context length:**
> | 370M↑ | 780M↑ | 1.4B↑ | 3B↑ |
> |-------|------|------|-----|
> | Mamba2 | 61.0 | 40.0 | 26.2 | 12.1 |
> | Mamba2 w/ full-attn | 59.4 | 38.8 | 26.1 | 12.5 |
> | RAMBA  | 57.3 | 37.5 | 25.5 | 11.1 |
>
> For short context lengths, the throughput of HSA is also comparable to both Mamba and Mamba with full attention.
> Fundamentally, the design of HSA does not introduce any drawbacks for short-text modeling.
>
> In summary, our experimental results cover context lengths from 64 to 32K and model parameters from 370M to 3B, and there is no objective evidence indicating any significant scalability deficiencies in HSA.
>
> Regarding design complexity, compared to the mainstream Mamba+attention approach, we replace full-attention with hierarchical sparse attention (HSA), with the only additional design being the chunk-wise encoder, whose necessity we have analyzed in the ablation study. Additionally, **memory reset is not part of the model architecture** but merely a training technique. Even without the memory reset, our model still outperforms other baselines. We aim to share more empirical insights with the community, even though it introduces some understanding overhead.
>
> As a scientific study, we strive to analyze issues based on objective data. We have provided all the supporting data for our claims and analyzed the necessity of each design component. If there are additional objective results that could address your remaining concerns, we welcome further discussion.

---

> ### Author Response · Authors · 2025-08-05
>
> Dear Reviewer fVSb
>
> As suggested by Reviewer XZCu, we consider moving the kernel design section to the appendix, freeing up space to provide more detailed explanations of our existing design rationales.
>
> **Why the chunk-wise Encoder is Necessary:**
> Each chunk requires a representation to be retrieved, which can be achieved through either a parameter-free or a parameterized approach. The parameter-free approach is mean pooling of hidden states or keys inside a chunk, while the parameterized approach introduces an additional encoder. We empirically find that using an extra encoder can significantly improve extrapolation performance. Thus, we included such a design in the paper along with an ablation study to validate its effectiveness.
>
> **Why Alternate One HSA Layer with G Mamba Layers in the Upper Decoder:**
> Current hybrid architectures combining Mamba and attention layers [1] typically alternate X layers of Mamba with one layer of attention. [1] indicates there's an optimal ratio exists, and excessive attention layers may lead to performance degradation under the same parameter budget. Therefore, our choice to alternate attention with multiple Mamba based on their empirical results.
>
> The modules we employ (Mamba, HSA, and chunk-wise encoder) are all essential. The proposed model architecture is a deliberate design supported by prior findings and our empirical results. We hope this clarification addresses concerns regarding complexity and provides a clearer understanding of the rationale behind our architecture.
>
> [1] An Empirical Study of Mamba-based Language Models

---

### Decision · Program_Chairs · 2025-09-17

**Decision:**

Accept (poster)

**Comment:**

The paper introduces Hierarchical Sparse Attention (HSA), designed to augment RNN architecture, with efficient long-range random access capabilities while preserving linear computational complexity and length generalization. The core claim is that HSA addresses a "trilemma" in long-context sequence modeling: balancing efficiency (linear time/space), random-access flexibility (ability to retrieve arbitrary historical tokens), and length extrapolation. HSA divides sequences into chunks, learns via a two-stage hierarchical process (intra-chunk encoding and inter-chunk aggregation using stick-breaking attention), and integrates hardware-aligned kernels, including KV cache offloading to CPU. When combined with Mamba-2, the resulting RAMba model is pretrained on 4K-length contexts but extrapolates to 64M tokens, achieving perfect accuracy on passkey retrieval tasks up to 64M, near-constant memory footprint during inference, and  comparable performance on benchmarks like RULER and LongBench. Reviewers characterize the work as a meaningful advance in chunk-based sparse attention.

Strengths:
1. A learnable two-stage sparse attention that directly addresses the efficiency–random access–length generalization trilemma. Overall the method is well-motivated and concretely presented.
2. Experimental results are solid.
3. Hardware-aligned kernels, and KV offload strategy makes it appliable

Weakness:
1. Generally, the proposed architectural is heavy in design, with HSA + chunk encoder + state passing + Mamba.
2. The focus is limited to Mamba, with no exploration of other RNNs or larger scales

I recommend acceptance as a poster. The key reasons include the paper's strong technical contributions to a high-impact area (long-context modeling in efficient architectures), backed by rigorous experiments and ablations.

The majority of reviewers (three out of four) rated it as accept or borderline accept, with high confidence, and the rebuttals effectively resolved most concerns through additional results.